# Spatial control of the APC/C ensures the rapid degradation of cyclin B1

Luca Cirillo[1,2], Rose Young [1,2], Sapthaswaran Veerapathiran[1], Annalisa Roberti [1], Molly Martin[1], Azzah Abubacar [1], Camilla Perosa[1], Catherine Coates [1], Reyhan Muhammad[1], Theodoros I Roumeliotis[1], Jyoti S Choudhary[1], Claudio Alfieri [1✉] & Jonathon Pines [1✉]

## Abstract

**The proper control of mitosis depends on the ubiquitin-mediated degradation of the right mitotic regulator at the right time. This is effected by the Anaphase Promoting Complex/Cyclosome (APC/C) ubiquitin ligase that is regulated by the Spindle Assembly Checkpoint (SAC). The SAC prevents the APC/C from recognising Cyclin B1, the essential anaphase and cytokinesis inhibitor, until all chromosomes are attached to the spindle. Once chromosomes are attached, Cyclin B1 is rapidly degraded to enable chromosome segregation and cytokinesis. We have a good understanding of how the SAC inhibits the APC/C, but relatively little is known about how the APC/C recognises Cyclin B1 as soon as the SAC is turned off. Here, by combining live-cell imaging, in vitro reconstitution biochemistry, and structural analysis by cryo-electron microscopy, we provide evidence that the rapid recognition of Cyclin B1 in metaphase requires spatial regulation of the APC/C. Using fluorescence cross-correlation spectroscopy, we find that Cyclin B1 and the APC/C primarily interact at the mitotic apparatus. We show that this is because Cyclin B1, like the APC/C, binds to nucleosomes, and identify an 'arginine-anchor' in the N-terminus as necessary and sufficient for binding to the nucleosome. Mutating the arginine anchor on Cyclin B1 reduces its interaction with the APC/C and delays its degradation: cells with the mutant, non-nucleosome-binding Cyclin B1 become aneuploid, demonstrating the physiological relevance of our findings. Together, our data demonstrate that mitotic chromosomes promote the efficient interaction between Cyclin B1 and the APC/C to ensure the timely degradation of Cyclin B1 and genomic stability.**

**Keywords** Mitosis; Cell Cycle; Ubiquitin; Chromosome; Nucleosome
**Subject Categories** Cell Cycle; Post-translational Modifications & Proteolysis

## Introduction

Protein degradation imposes unidirectionality on the cell cycle through the rapid degradation of key mitotic regulators (Cyclin B and Securin) in metaphase. This ensures that chromosome segregation and mitotic exit are rapid and effectively irreversible, but the corollary is that to maintain genome stability, degradation must be coordinated with chromosome attachment to the mitotic apparatus. To achieve this, the spindle assembly checkpoint (SAC) monitors kinetochore attachment to microtubules and prevents the anaphase-promoting complex/cyclosome (APC/C) ubiquitin ligase from recognising Cyclin B and Securin until all chromosomes have attached to the spindle (reviewed in Alfieri et al, 2017; Yamano, 2019). Once all the chromosomes have attached to the spindle, the APC/C rapidly recognises Cyclin B1 and Securin to drive sister chromatid separation and exit from mitosis. Although we know many of the components of the machinery and their molecular structures, we lack an understanding of how they interact in living cells to generate rapid and responsive pathways.

The APC/C is activated at nuclear envelope breakdown by phosphorylation of the APC/C subunits APC1 and APC3 and dephosphorylation of its co-activator, CDC20 (Fujimitsu et al, 2016; Fujimitsu & Yamano, 2021; Kraft et al, 2003; Labit et al, 2012; Steen et al, 2008; Yamano, 2019; Zhang et al, 2016). This enables CDC20 to bind and activate the apo-APC/C, but it is then kept in check by the SAC. The SAC inhibits the APC/C by generating the Mitotic Checkpoint Complex (MCC, reviewed in Lara-Gonzalez et al, 2021; McAinsh and Kops, 2023), that binds tightly to the APC/C as a pseudo-substrate inhibitor (Alfieri et al, 2016; Izawa and Pines, 2015). When bound to the MCC, APC/C is unable to recognise and ubiquitylate Cyclin B1 and Securin (Clute & Pines, 1999; Hagting et al, 2002; Heasley et al, 2017; Hwang et al, 1998; Kim et al, 1998; Rieder et al, 1997; Thornton & Toczyski, 2003) although it is still able to recognise early targets such as Cyclin A2 and NEK2A (Alfieri et al, 2020; den Elzen & Pines, 2001; Di Fiore & Pines, 2010; Geley et al, 2001; Hames et al, 2001, 2001; Zhang et al, 2019). Once all chromosomes are properly attached to spindle microtubules, the SAC is silenced and the APC/C can ubiquitylate Cyclin B1 and Securin, whose proteolysis activates the Separase protease (Holland & Taylor, 2006; Hornig et al, 2002; Salah &

[1]The Institute of Cancer Research Chester Beatty Laboratories, 237 Fulham Road, London SW3 6JB, UK. [2]These authors contributed equally: Luca Cirillo, Rose Young.
✉E-mail: claudio.alfieri@icr.ac.uk; jon.pines@icr.ac.uk

Nasmyth, 2000; Stemmann et al, 2001; Wirth et al, 2006; Yu et al, 2021) to cleave cohesin and allow sister chromatids to separate.

Strict temporal inhibition of the APC/C is necessary to prevent premature chromosome separation that may lead to the gain or loss of chromosomes by the daughter cells, but single-cell imaging has shown that Cyclin B1 rapidly begins to be degraded once all the chromosomes are attached to the spindle (Jackman et al, 2020), which ensures rapid exit from mitosis and could conceivably be required for the fidelity of chromosome segregation. How the APC/C rapidly recognises its substrates when the SAC is silenced is not understood. One potential clue comes from our previous work showing that in HeLa cells Cyclin B1-GFP disappears first from chromosomes and centrosomes compared with the cytoplasm (Clute & Pines, 1999). Moreover, fractionation experiments on HeLa cells showed that the APC/C may be more active on chromatin (Sivakumar et al, 2014). In Drosophila, Cyclin B1-GFP first disappears from centrosomes and then from the spindle and the chromosomes (Huang & Raff, 1999). These observations indicate that the APC/C might be spatially regulated in mitosis, which could ensure that Cyclin B1 is removed first from specific locations to coordinate the events of anaphase and mitotic exit. Yet despite these indications, direct evidence for spatial regulation of the APC/C has been lacking.

Here, we provide direct evidence for the spatial control of APC/C activity. Using live-cell imaging and fluorescence cross-correlation spectroscopy (FCCS), we have assayed the binding between Cyclin B1 and the APC/C and found that the spatial pattern of their interaction correlates with that of Cyclin B1 degradation. Moreover, we have identified the molecular mechanism promoting this localised interaction: we found that a loop on APC3 and the N-terminus of Cyclin B1 both contain distinct nucleosome-binding motifs that regulate chromatin localisation. Preventing Cyclin B1 from binding to chromatin delays its degradation and cells become aneuploid; timely degradation can be restored by recruiting a non-nucleosome-binding Cyclin B1 mutant back to chromatin. We conclude that nucleosomes facilitate the interaction between Cyclin B1 and the APC/C to promote the rapid degradation of Cyclin B1 during metaphase and safeguard genomic stability.

# Results

## Cyclin B1 degradation is spatially regulated during metaphase

We used CRISPR/Cas9 gene editing to tag both alleles of Cyclin B1 in human retinal pigment epithelial (RPE-1) cells with the mEmerald fluorescent protein (Barbiero et al, 2022; Cubitt et al, 1998). As we and others previously reported, live-cell imaging revealed that Cyclin B1 localised to specific structures in mitotic cells, including the spindle microtubules, spindle caps, and chromosomes (Bancroft et al, 2020; Bentley et al, 2007; Huang and Raff, 1999; Jackman et al, 1995; Pines and Hunter, 1991), and Cyclin B1 fluorescence disappeared rapidly at the start of metaphase when the SAC is turned off (Clute & Pines, 1999; Hagting et al, 1998; Jackman et al, 2020, 2003). In agreement with our previous results using ectopically expressed Cyclin B1-FP (Clute & Pines, 1999), measuring the fluorescence intensity of

Cyclin B1 at different subcellular locations showed that Cyclin B1 disappeared first from the chromosomes, spindle and centrosomes, and only later in the cytoplasm (Fig. 1A,B; Movie EV1).

Two non-exclusive mechanisms could account for the faster disappearance of Cyclin B1 in specific places: Cyclin B1 could be rapidly displaced, or it could be targeted for degradation in situ. To distinguish between these two possibilities, we measured the disappearance of Cyclin B1 fluorescence in cells treated either with MG132 to inhibit the proteasome or with a combination of the APC/C inhibitors APCin and proTame (Sackton et al, 2014; Verma et al, 2004) to inhibit ubiquitylation. Both treatments stabilised Cyclin B1 levels and arrested cells in metaphase as previously reported (Sackton et al, 2014; Zeng et al, 2010), and Cyclin B1 levels remained constant at all the subcellular locations tested (Fig. 1C,D). This indicated that the pattern of Cyclin B1 disappearance in metaphase is linked to its ubiquitin-mediated degradation and not to its rapid displacement.

## Analysing Cyclin B1–APC/C interaction by fluorescence cross-correlation spectroscopy

Our results indicated that the proteolysis of endogenous Cyclin B1 is spatially regulated, therefore we set out to determine the mechanism responsible. This could originate from the differential activity of the APC/C, the proteasome, or both. The APC/C was previously reported to be more active at centrosomes (Kraft et al, 2003) or chromosomes (Sivakumar et al, 2014; Topper et al, 2002); therefore, we focused first on the interaction between Cyclin B1 and the APC/C. To measure protein-protein interactions with subcellular resolution in living cells, we turned to fluorescence cross-correlation spectroscopy (FCCS), which we recently validated as a tool to study the cell cycle machinery (Barbiero et al, 2022). Since FCCS requires that the entire population of proteins under investigation be fluorescent, we used CRISPR/Cas9$^{D10A}$ gene-editing (Shen et al, 2014) to introduce the mScarlet (Bindels et al, 2017) fluorescent protein into both alleles of the APC/C subunit APC8 in our RPE-1 Cyclin B1-mEmerald$^{+/+}$ cells and the parental RPE-1 cells. PCR and sequencing confirmed biallelic insertion in three independent clones of the parental RPE-1 cells and of the RPE-1 Cyclin B1-mEmerald$^{+/+}$ cells (see source data).

We first characterised the APC8-mScarlet cells to determine whether the tag perturbed the APC/C function. APC8-mScarlet was enriched in the nucleus of interphase cells; during mitosis, it was weakly enriched at centrosomes but otherwise homogeneous throughout the cell (Figs. 1E and Fig. EV1A,B), in agreement with previous immunofluorescence data of other APC/C subunits (Acquaviva & Pines, 2006; Huang & Raff, 2002; Kraft et al, 2003; Tischer et al, 2022; Tugendreich et al, 1995). The mScarlet tag did not significantly alter mitotic timing in either unperturbed mitosis or in cells treated with low doses of the microtubule poison paclitaxel to prolong the SAC (Fig. EV1C). Similarly, tagging APC8 had no significant effect on cell growth rate or ploidy (Fig. EV1D). Although immunoblotting indicated that the protein levels of APC8-mScarlet were about 70% lower than wild-type APC8 in parental cells (Fig. EV1E,F), this did not significantly affect the rate of Cyclin B1-mEmerald degradation in mitosis (Fig. EV1H). The mScarlet tag did not interfere with APC8 incorporation into the APC/C because when we immunodepleted APC4 there was no difference in the amount of APC8 or APC8-mScarlet remaining in

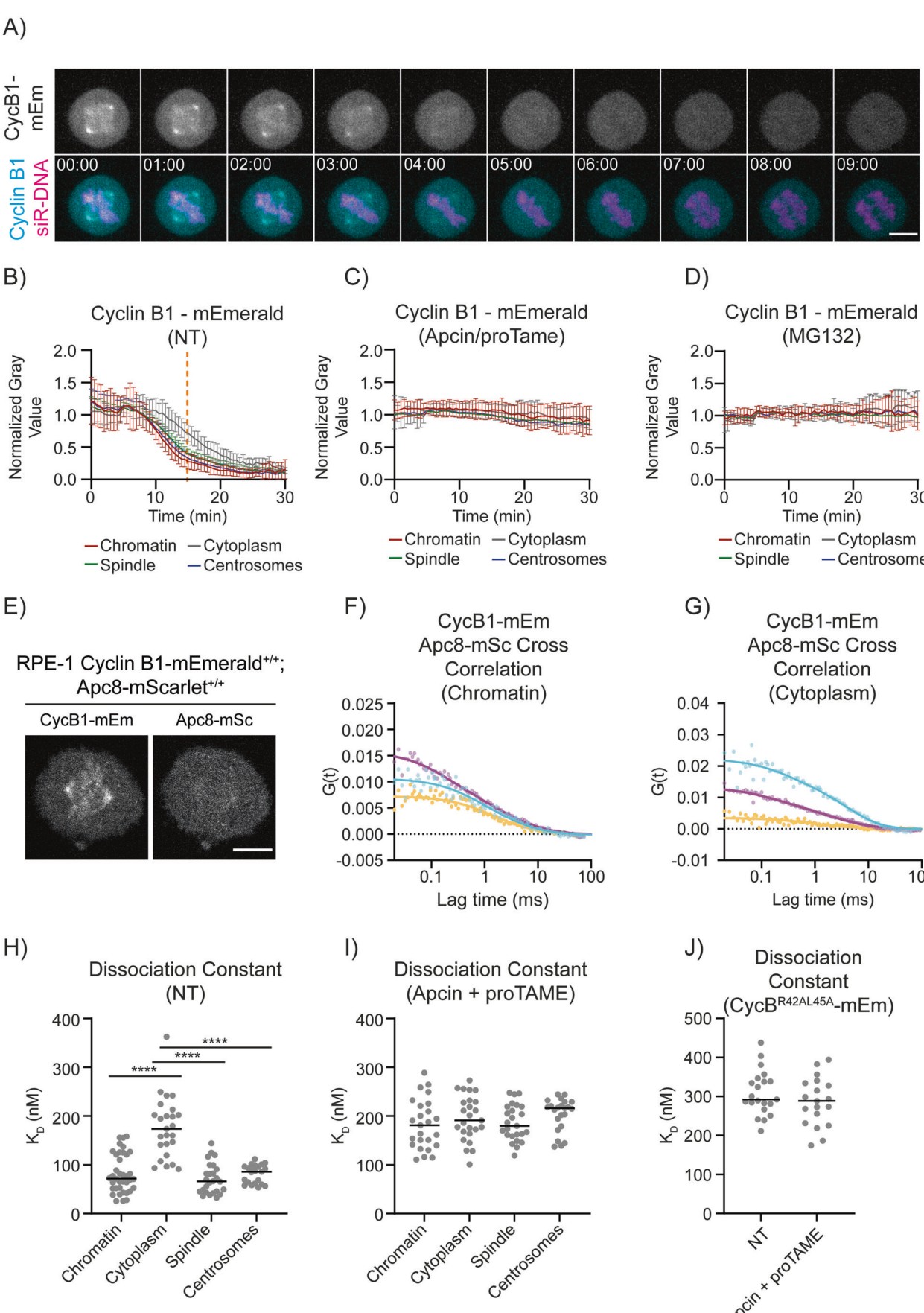

◄  **Figure 1.  Cyclin B1 degradation is spatially regulated.**

(**A**) Representative maximum projections of fluorescence confocal images over time of an RPE-1 Cyclin B1-mEmerald$^{+/+}$ cell progressing through mitosis. Time is expressed as mm:ss. The scale bar corresponds to 10 μm. (**B–D**) Quantification of Cyclin B1 fluorescence levels (normalised raw integrated density, RID) over time in RPE-1 Cyclin B1-mEmerald$^{+/+}$ cells: $n \geq 18$ cells per condition, $N = 3$ independent experiments. Mean ± standard deviation are plotted. (**E**) Representative fluorescence confocal images of RPE-1 Cyclin B1-mEmerald$^{+/+}$; APC8-mScarlet$^{+/+}$ cells. The scale bar corresponds to 10 μm. (**F, G**) Representative graph of the autocorrelation of Cyclin B1-mEmerald and APC8-mScarlet and the cross-correlation between the two at the chromatin (**F**) and in the cytoplasm (**G**). In this and all following FCCS graphs, dots = data, lines = fit; Cyan = Cyclin B1-mEmerald, magenta = APC8-mScarlet, yellow = cross-correlation between cyclin B1-mEmerald and APC8-mScarlet. (**H–J**) Dot plots representing the $K_D$ between endogenous Cyclin B1-mEmerald and APC8-mScarlet in the indicated conditions: $n \geq 21$ cells per condition, $N = 3$ independent experiments. In this and all following $K_D$ dot plots, each dot corresponds to a single measurement, horizontal black lines represent median values. In this and all following figures, the orange dotted line indicates anaphase (panel **B**), the dotted horizontal line indicates 0 (panels **F**, **G**), CycB1 Cyclin B1, mEm mEmerald, mSc mScarlet, see Appendix Table S2 for statistical tests and *p* values. ns not significant, $p > 0.05$; *$p < 0.05$, **$p < 0.01$, ***$p < 0.001$, ****$p < 0.0001$ for all figures. Source data are available online for this figure.

the supernatant (Fig. EV1E,F—see additional text for Fig. EV1). In agreement with this, FCS analysis of APC8-mScarlet in interphase and mitosis indicated that APC8-mScarlet behaved as a single, slow diffusing species with a diffusion coefficient of $6.44 \pm 1.79\ \mu m^2\,s^{-1}$ (Mean ± SD) and hydrodynamic radius of ~14 nm, which is comparable to the theoretical hydrodynamic radius of APC/C (Fig. EV1G, Alfieri et al, 2016; Chang et al, 2015; Yamaguchi et al, 2016). We concluded that APC8 tagged with mScarlet was a valid reporter for the APC/C.

## D-box dependent Cyclin B1–APC/C interaction is favoured on chromosomes and in the spindle

Validated cell lines where both alleles of cyclin B1 and APC8 were tagged with fluorescent proteins enabled us to use FCCS to study their interaction during cell division (Fig. 1E). In agreement with its SAC-dependent degradation, there was no cross-correlation between Cyclin B1 and the APC/C in G2 phase cells but a strong interaction in metaphase cells (Figs. 1F,G and EV2A; Appendix Table S1). We used these data to calculate the dissociation constant ($K_D$) between Cyclin B1-mEmerald and APC8-mScarlet at different subcellular locations and found that Cyclin B1 and the APC/C bound more strongly (had a lower $K_D$) at the chromatin, spindle and spindle poles than in the cytoplasm (Figs. 1F–H and EV2B,C). We obtained similar results in cells arrested in metaphase with the proteasome inhibitor MG132 (Fig. EV2D). To validate our FCCS assay, we measured cross-correlation in the presence of the small molecule inhibitors APCin and proTAME that impair CDC20 substrate recognition and CDC20 binding to the APC/C, respectively (Sackton et al, 2014). In the presence of APCin and proTAME, the $K_D$ between Cyclin B1-mEmerald and APC8-mScarlet in the spindle, on chromosomes and centrosomes (Fig. 1I) increased to a level similar to that found in the cytoplasm of either treated or untreated cells (Fig. 1I).

The residual cross-correlation in the presence of APCin and proTAME indicated that there was a second, lower affinity mode of binding between Cyclin B1 and the APC/C, which was consistent with Cyclin B1-Cdk1-Cks binding to phospho-APC/C through the Cks protein (Kraft et al, 2003; Patra & Dunphy, 1996; Rudner & Murray, 2000; Shteinberg & Hershko, 1999; Sudakin et al, 1997; van Zon et al, 2010). Congruent with this, there was low-level cross-correlation between Cyclin B1-mEmerald and APC8-mScarlet in prometaphase cells (Fig. EV2E,F), and this interaction was independent of the Cyclin B1 D-box (Figs. 1J and EV2G) and unaffected by treatment with APCin and proTAME (Fig. 1J).

We concluded that FCCS revealed two major modes of binding between Cyclin B1 and the APC/C in dividing RPE-1 cells: a high

affinity, D-box-dependent interaction spatially constrained to the spindle and chromosomes and temporally restricted to metaphase; plus a D-box-independent, lower affinity interaction predominantly in the cytoplasm in both prometaphase and metaphase cells that was likely mediated by Cks. At present, we cannot definitively conclude that there is no D-box-dependent binding in the cytoplasm because there could be a population too small to distinguish by FCCS.

## Chromatin is the major site of Cyclin B1 disappearance in human cells

Our data indicated that Cyclin B1 interaction with the APC/C was favoured on the chromosomes, spindle and centrosomes, but the relative contribution of each subcellular location to Cyclin B1 degradation was unclear. To answer this, we asked whether Cyclin B1 would disappear faster on chromosomes spatially separated from the metaphase plate. We generated cells in which some chromosomes remained at the spindle poles by inhibiting the CENP-E (Centromeric Protein E) kinesin motor protein that is required for these chromosomes to congress to the metaphase plate (Fig. 2A, Bennett et al, 2015). If individual chromosomes are important for Cyclin B1 degradation, then the degradation of Cyclin B1 should be faster on polar chromosomes than in the surrounding cytoplasm. Cells treated with a CENP-E inhibitor arrested in mitosis with polar chromosomes and stable levels of Cyclin B1 (Fig. 2B). When we treated these cells with the MPS1 inhibitor Reversine to inactivate the SAC (Chen et al, 2004), Cyclin B1 disappeared earlier from polar chromosomes than it did from the surrounding cytoplasm (Fig. 2B; Movie EV2). The decay of Cyclin B1 at polar chromatids was indistinguishable from that of aligned chromosomes.

This result demonstrated that the chromosomes themselves, rather than their relative position in the cell, facilitated Cyclin B1 degradation in metaphase, and chromosomes likely represent a major site of Cyclin B1 degradation.

## The APC3 unstructured loop directly binds the nucleosome acidic patch

Our data supported the conclusion that chromosomes are the preferential site of Cyclin B1 interaction with the APC/C. A previous large-scale study found that the APC/C associates with nucleosomes via their acidic patch (Skrajna et al, 2020), which could be relevant to how chromosomes facilitate Cyclin B1 degradation. We used an Electrophoretic Mobility Shift Assay (EMSA) to test whether a complex composed of APC/C, CDC20

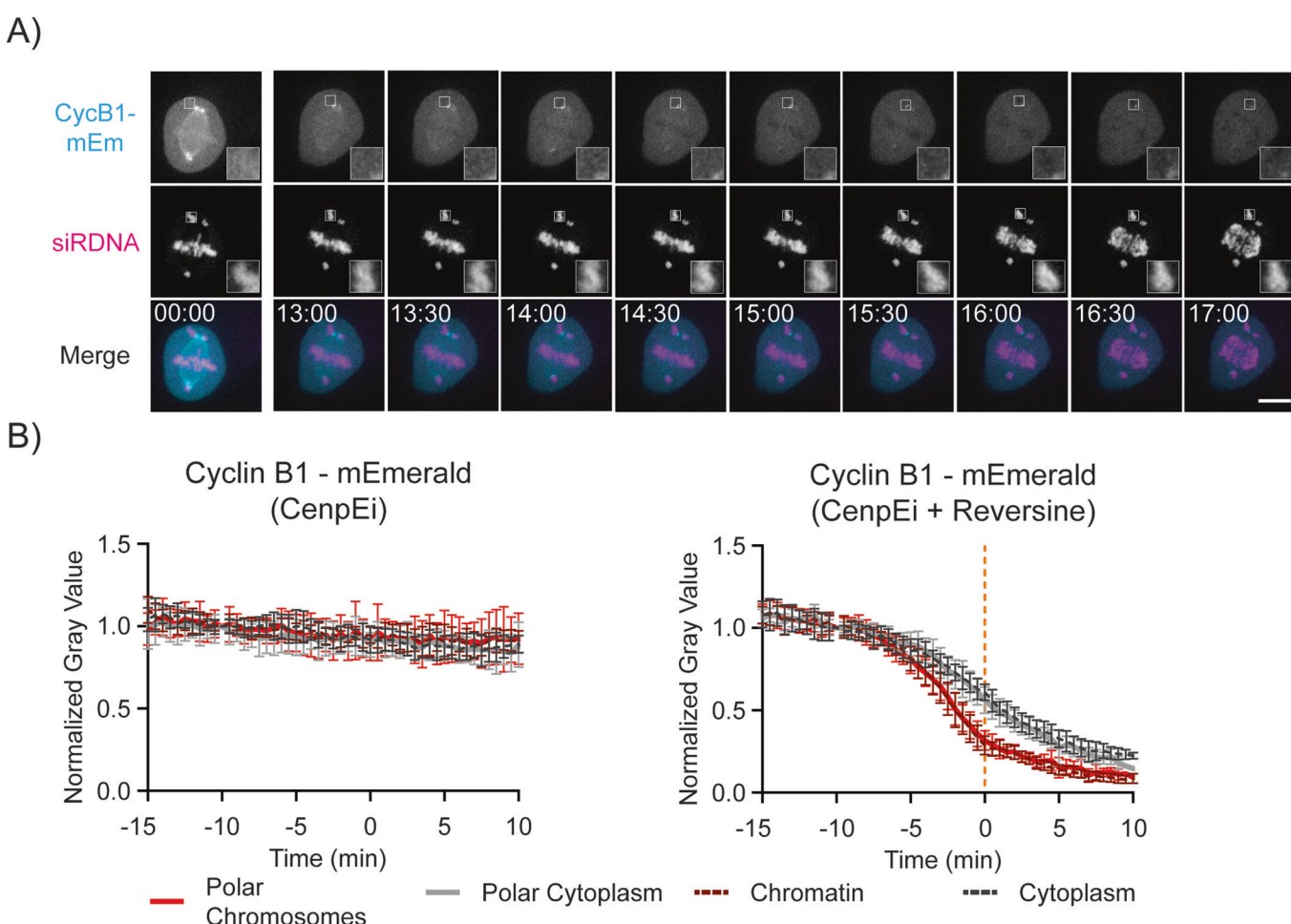

**Figure 2. Chromatin is a major site for Cyclin B1 disappearance in RPE-1 cells.**

(A) Representative fluorescence confocal images over time of an RPE-1 Cyclin B1-mEmerald[+/+] cell progressing through mitosis after treatment with CenpEi and Reversine. Time is expressed as mm:ss. The scale bar corresponds to 10 μm. (B) Quantification of normalised Cyclin B1 fluorescence levels over time in RPE-1 Cyclin B1-mEmerald[+/+] cells after treatment with CenpEi inhibitor (left) or a combination of CenpEi and Reversine (Right): $n = 11$ cells per condition, $N = 3$ independent experiments. Mean ± standard deviation are plotted. Source data are available online for this figure.

and an N-terminal fragment of Cyclin B1 (Cyclin B1[NTD]) could bind Nucleosome Core Particles (NCP). We found that increasing concentrations of APC/C-CDC20-Cyclin B1[NTD] shifted the NCP towards higher molecular weights, indicating that APC/C-CDC20-Cyclin B1[NTD] bound to the NCP (Figs. 3A and EV2H). To determine how this complex bound to nucleosomes, we used Cross-Linking mass spectrometry (XL-MS). In addition to the expected crosslinks within the APC/C-CDC20 complex and within the NCP (Fig. EV3A,B), we identified two crosslinks from the APC/C to the NCP (Fig. 3B,C): K390 from the APC3 unstructured loop crosslinked to residues 96 and 106 of H2A and H2B, respectively, which are proximal to the acidic patch of the nucleosome. The same interaction was detected when we performed XL-MS using only APC/C and NCP (Fig. EV3C,D). To refine this analysis, we expressed and purified the APC3 unstructured loop in baculovirus and found that this was sufficient to shift the NCP in an EMSA (Fig. 3D). K390 lies proximal to arginine residues that conform to an arginine anchor motif that is used by several proteins to bind to the nucleosome acidic patch (reviewed in Kalashnikova et al, 2013;

Paul, 2021), and mutating them to glutamic acid (APC3 loop[3R3E]) abolished nucleosome binding in our EMSA assay (Fig. 3D). (Note that the mutant APC3 loop still binds non-specifically to nucleosomes, which accounts for the disappearance of the NCP147 band but does not result in a well-defined shifted product.) In addition, incubating the NCP and the APC3 loop with a peptide that binds strongly to the acidic patch (the LANA - Latency Associated Nuclear Antigen - peptide of Kaposi Sarcoma Herpes Virus, Barbera et al, 2006) prevents the formation of the APC3 loop[3R3E]:NCP complex (Fig. 3E).

To visualise the interaction surfaces between APC3 and the NCP, we determined the cryo-electron microscopy (cryo-EM) structure of the APC3 loop–NCP complex at 2.5 Å resolution (Appendix Table S3; Fig. EV4). The excellent quality of the map allowed unambiguous ab initio model building of APC3 loop residues 375–381 (containing the sequence PRRSSRL) that bind the NCP acidic patch. R380 of the APC3 loop is the arginine anchor that directly interacts with acidic patch residues D90 and E92 (Fig. 3F,G). To the N-terminus, the peptide forms a saddle on top

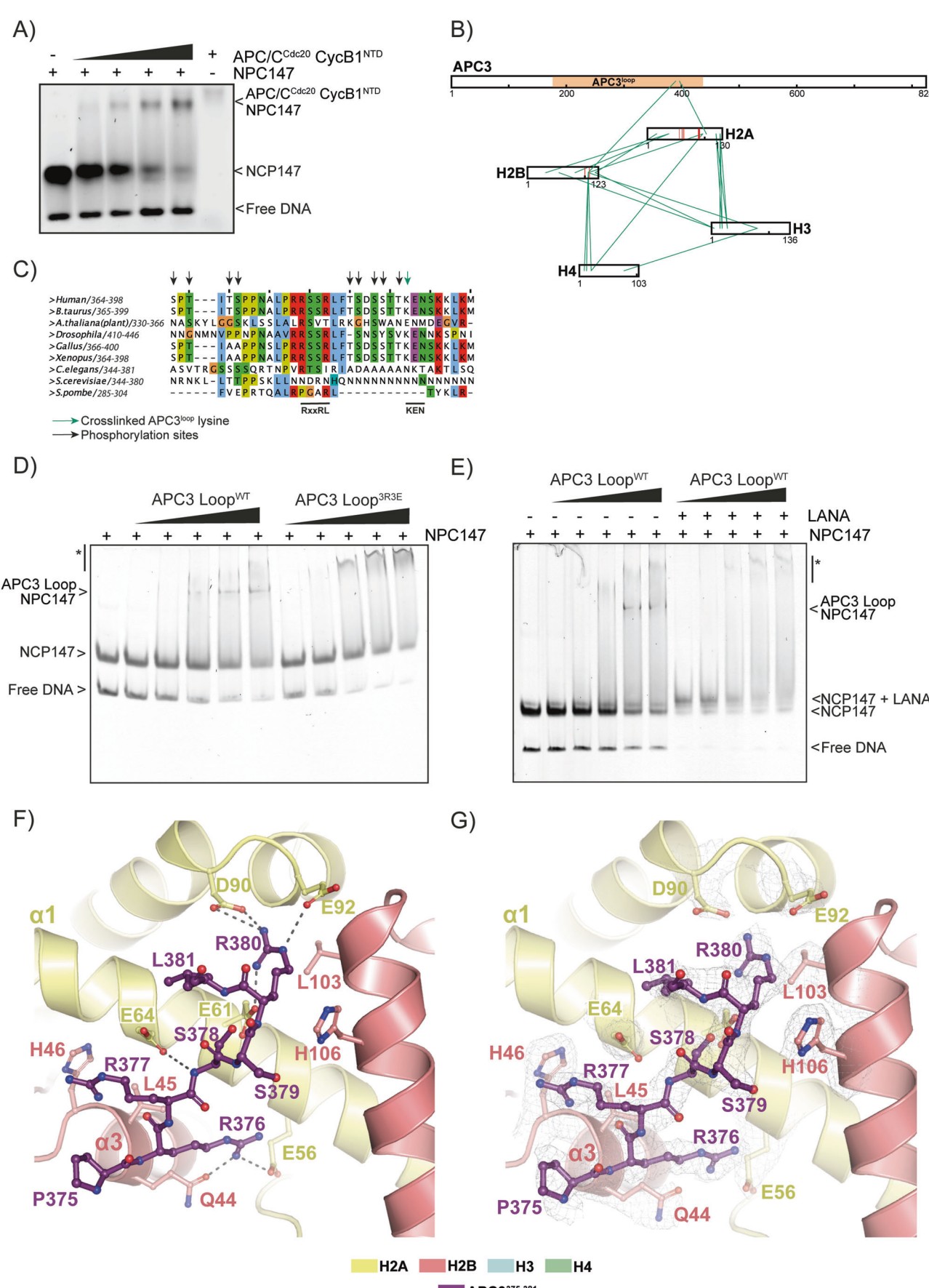

**Figure 3. The APC3 loop of APC/C binds the nucleosome acidic patch.**

(A) EMSA of the APC/C-CDC20-Cyclin B1$^{NTD}$ and NCP147. $N = 3$ independent experiments. (B) Representation of the heteromeric crosslinks found between the APC3 loop and the acidic patch of the NCP. (C) A sequence alignment of the C-terminus of the human APC3 loop with homologous sequences in other species. APC3 K390, which crosslinks with the acidic patch, is marked with a green arrow. Previously identified phosphorylation sites are marked with black arrows. (D) EMSA of the APC3 loop WT and charge-substitution 3R3E mutant with the NCP147. $N = 3$ independent experiments. Bands corresponding to free nucleosome and to the APC3:nucleosome complex are marked. Non-specific binding events are indicated with an asterisk. (E) EMSA of the APC3 loop WT including LANA peptide. $N = 3$ independent experiments. (F) A close-up view of the APC3 loop RRSSR motif bound to the acidic patch of the NCP. The APC3 loop is shown as balls and sticks in dark purple, residues 375–381 can be seen. H2A and H2B are shown in the cartoon representation. Dashed lines indicate hydrogen bonds between the APC3 loop and the acidic patch and surrounding residues. R380 establishes hydrogen bonds with E61, D90, and E92 of the acidic patch, and R376 with E56. (G) Cryo-EM density of the APC3 loop and surrounding H2A/ H2B residues. Source data are available online for this figure.

of H2A helix α3, thereby reaching H2B helix α1 where Q44 and E56 of H2A both interact with APC3 R376. The interaction is stabilised further by H2A E64 that interacts with the peptide backbone of the APC3 loop (Fig. 3F,G).

We conclude that the APC/C binds the nucleosome through an arginine anchor motif in the APC3 loop that engages the nucleosome on a surface encompassing the acidic patch.

## The N-terminus of Cyclin B1 directly interacts with the nucleosome acidic patch

Having established how the APC/C binds nucleosomes, we investigated the interaction between Cyclin B1 and chromosomes. We focused on the N-terminus of Cyclin B1 because we had shown that the N-terminal nine amino acids of Cyclin B1 are required for rapid degradation (Matsusaka et al, 2014), and work from the King lab demonstrated that arginines in the N-terminus of Cyclin B1 were important for chromosome recruitment (Bentley et al, 2007; Pfaff and King, 2013).

The sequence of the first seven amino acids of Cyclin B1 are conserved and resembles an arginine anchor motif (Fig. 4A,B). We used an EMSA to test the possibility that Cyclin B1 could bind directly to nucleosomes (Figs. 4C and EV3E). Adding increasing amounts of Cyclin B1 to the nucleosomes shifted the DNA-nucleosome band towards higher molecular masses, indicating the formation of a complex (Figs. 4C and EV3E). We obtained similar results using an N-terminal peptide containing the first 23 amino acids of Cyclin B1 (Fig. EV3F). We tested whether this interaction was mediated by the putative arginine anchor in the amino terminus of Cyclin B1 by mutating the arginine residues and observed a significant decrease in the binding affinity in charge-substitution mutants (Cyclin B1$^{4E7E}$, Figs. 4C and EV3E), or when the first nine amino acids of Cyclin B1 were deleted (Cyclin B1$^{Δ9}$), or when amino acids 3 to 7 were mutated to alanine (Cyclin B1$^{3A5}$, Fig. EV3G,H).

To gain further insight into the mode of binding of Cyclin B1 to the nucleosome, we employed cryo-EM to determine the structure of NCP bound to a peptide containing the first 21 residues of Cyclin B1 (NCP$^{CbNT}$). We obtained a reconstruction at 2.5 Å resolution (Appendix Table S3) showing residues 2–4 of Cyclin B1 interacting with the nucleosome acidic patch on the NCP (Fig. 4D). Cyclin B1 Arginine 4 establishes hydrogen bonding with the acidic patch residues Aspartic acid 90 and Glutamic acid 92 in H2A (Figs. 4E–G and EV5), and the peptide backbone of Cyclin B1 Alanine 2 interacts with H2B Glutamic acid 110.

To determine whether the N-terminus of Cyclin B1 was required for its chromosomal localisation in living cells, we expressed tetracycline-inducible Cyclin B1-mScarlet variants in the background of our RPE-1 Cyclin B1-mEmerald$^{+/+}$ cells. We reasoned that by comparing the fluorescence intensity of mEmerald and mScarlet we would be able to assess the localisation of ectopically expressed Cyclin B1 using endogenous Cyclin B1 as an internal control (Fig. 4H). To facilitate the analysis, we labelled DNA with siR-DNA and treated cells with the proteasome inhibitor MG132 to induce a metaphase arrest (Potapova et al, 2006). By measuring the fluorescence intensity along a line drawn from centrosome to centrosome, we observed a three-peak pattern for Cyclin B1-mEmerald whereby the first and the last peaks correspond to the two centrosomes and the middle peak colocalised with the maximum DNA signal (Fig. 4I; Appendix Fig. S1A). Ectopic Cyclin B1$^{WT}$-mScarlet largely overlapped with endogenous Cyclin B1-mEmerald, whereas a Cyclin B1$^{4E7E}$-mScarlet mutant failed to localise to the chromosomes (Fig. 4H,I; Appendix Fig. S1A). We obtained comparable results by deleting the first 9 residues of CyclinB1, or by substituting residues 3 to 7 for alanine (Appendix Fig. S1A). To determine whether the N-terminus of Cyclin B1 was sufficient to localise a protein to the chromosomes, we tagged Securin with residues 1 to 9 of Cyclin B1. Securin-mScarlet, was enriched at the mitotic spindle but largely excluded from chromatin (Hagting et al, 2002), but the Cyclin B1-Securin fusion protein (Cyclin B1$^{(1-9)}$-Securin-mScarlet) partially localised to chromosomes (Fig. 4J; Appendix Fig. S1B–D). Fusing Securin-mScarlet to the LANA peptide as a positive control strongly enriched Securin on chromosomes (Fig. 4J; Appendix Fig. S1C).

We conclude that the N-terminus of Cyclin B1 acts as an arginine anchor to dock Cyclin B1 to the acidic patch of nucleosomes during mitosis.

## Nucleosome binding is important for timely Cyclin B1 degradation

We previously showed that the first 9 residues of Cyclin B1 are important for timely Cyclin B1 degradation (Matsusaka et al, 2014), providing a potential link between the localisation of Cyclin B1 at chromatin and its degradation. To investigate whether nucleosome binding is required for the timely degradation of Cyclin B1, we compared the degradation of endogenous Cyclin B1-mEmerald with that of ectopically expressed Cyclin B1-mScarlet variants. While Cyclin B1$^{WT}$-mScarlet was degraded slightly before endogenous Cyclin B1, mutants preventing acidic patch binding— Cyclin B1$^{3A5}$-mScarlet, Cyclin B1$^{Δ2-9}$-mScarlet and Cyclin B1$^{4E7E}$-mScarlet—were all degraded later (Fig. 5A,B; Appendix Fig. S2A). When we measured the interaction between the APC/C and

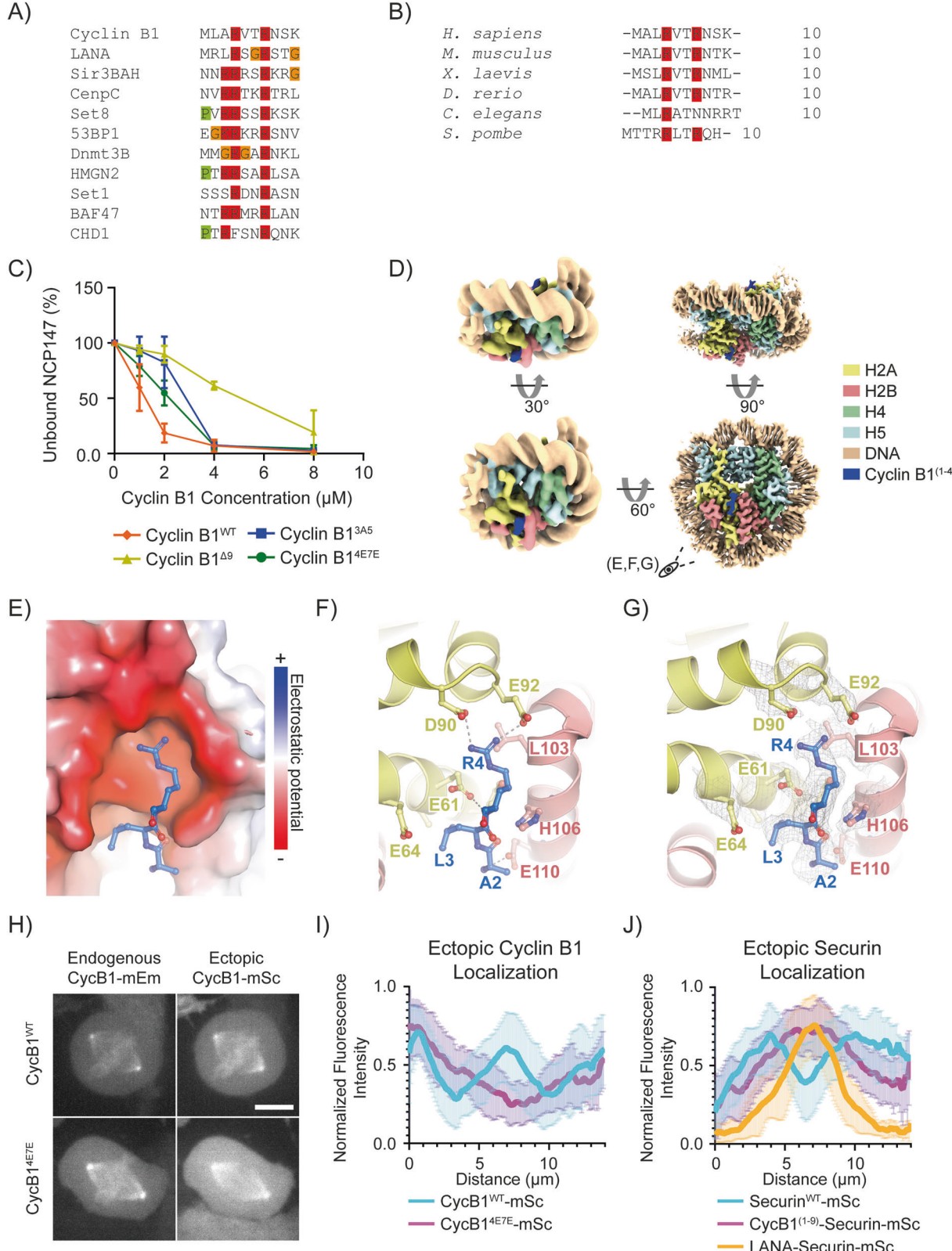

**Figure 4. Cyclin B1 N-terminus mediates nucleosome binding.**

(A) Protein alignment of the N-terminus of human Cyclin B1 with known arginine anchors of other nucleosome interacting proteins. (B) Protein alignment of the N-terminus of Cyclin B1 orthologues. (C) Quantification of EMSAs of full-length Cyclin B1$^{WT}$ and the indicated Cyclin B1 variants. $N = 3$ independent experiments. Mean ± standard deviation are plotted. (D) Cryo-EM structure of the NCP in complex with Cyclin B N-terminus (NCP$^{CbNT}$). On the left, the cryo-EM density is low-pass filtered to 7 Å to show the nucleosome particle in its entirety (including the less rigid entry and exit DNA). On the right, the same structure is shown at full power (2.5 Å resolution). (E) A close-up view of D, residues 2–4 of Cyclin B1 interact with the acidic patch of NCP147. Cyclin B1 is shown as a ball and stick, histones H2A and H2B are shown as electrostatic surface potentials ($-/+5.000$). (F) Same structure as in (E), however, H2A and H2B histones are shown as ribbon models. Interacting side chains are depicted. Dashed lines indicate hydrogen bonds between R4 of Cyclin B1 and acidic patch residues D90 and E92. Cyclin B1 peptide backbone at A2 interacts with E110 of H2B. (G) Cryo-EM density on the side chains shown in (F). (H) Maximum projections of confocal images representative of RPE-1 Cyclin B1-mEmerald$^{+/+}$ cells ectopically expressing the indicated variant of Cyclin B1-mScarlet. The scale bar represents 10 μm. (I) Line-profile graph representing the pixel-by-pixel fluorescence intensity over a line drawn from centrosome to centrosome of RPE-1 Cyclin B1-mEmerald$^{+/+}$ cells ectopically expressing the indicated Cyclin B1 variant: $n = 18$ cells per condition, $N = 3$ independent experiments. Mean ± standard deviation are plotted. (J) Line-profile graph representing the pixel-by-pixel fluorescence intensity over a line drawn from centrosome to centrosome of RPE-1 Cyclin B1-mEmerald$^{+/+}$ cells ectopically expressing the indicated Securin variant: $n \geq 16$ cells per condition, $N = 3$ independent experiments. Mean ± standard deviation are plotted. Source data are available online for this figure.

ectopically expressed Cyclin B1 using FCCS we found that—like endogenous Cyclin B1—the cross correlation between Cyclin B1$^{WT}$-mEmerald and APC8-mScarlet was stronger on the chromosomes than in the cytoplasm (Fig. 5C; Appendix Fig. S2B,C), whereas Cyclin B1$^{4E7E}$-mEmerald only cross-correlated with APC8-mScarlet in the cytoplasm (Fig. 5D; Appendix Fig. S2C,D). The cytoplasmic cross correlation between Cyclin B1$^{4E7E}$-mEmerald and APC8-mScarlet remained unchanged after treating cells with APCin and proTAME, indicating that this interaction was not mediated by the D-box (Appendix Fig. S2D).

The delay in the degradation of Cyclin B1 mutants indicated that chromatin localisation might be important for Cyclin B1 degradation, but an alternative explanation could have been that the N-terminus of Cyclin B1 acted as an additional degron to enhance binding to the APC/C. To test this, we assayed Cyclin B1$^{\Delta 9}$ binding to APC/C using size-exclusion chromatography (Fig. 6A,B) and found that purified Cyclin B1 and Cyclin B1$^{\Delta 9}$ co-eluted identically with the APC/C. This indicated that the arginine anchor motif of Cyclin B1 did not measurably contribute to APC/C binding.

If chromatin localisation is important for Cyclin B1 degradation, we reasoned that restoring the localisation of Cyclin B1$^{\Delta 9}$ should restore timely degradation. We tested this by fusing the LANA peptide to Cyclin B1$^{\Delta 9}$. In agreement with our hypothesis, the LANA-Cyclin B1$^{\Delta 9}$-mScarlet fusion protein was highly enriched at chromatin and its degradation rate was similar to that of endogenous Cyclin B1 (Fig. 6C). (Note that it was immaterial whether we fused the LANA peptide to the N or C terminus of the Cyclin (Appendix Fig. S3A). Moreover, FCCS measurements confirmed that ectopically expressed LANA-Cyclin B1$^{\Delta 9}$-mEmerald cross-correlated with endogenous APC8-mScarlet at the chromosomes (Appendix Fig. S3B,C). A control fusion between Cyclin B1$^{\Delta 9}$ and a mutant of LANA unable to bind nucleosomes failed both to localise to chromatin and to restore Cyclin B1 degradation (Fig. 6D).

Our results raised the question of whether localising Securin, the other metaphase APC/C substrate, to chromatin would be sufficient to enhance its degradation. Indeed, we found that ectopically expressed LANA-Securin-mScarlet was highly enriched at chromatin (Fig. 4J) and its degradation began earlier than wild-type Securin-mScarlet (Appendix Fig. S4A,B). We also observed a slight advance in Securin degradation when it was fused to the first nine residues of Cyclin B1 (Appendix Fig. S4A,B). Overexpressing Securin did not affect the subcellular degradation of Cyclin B1, indicating that competition between APC/C substrates (Kamenz

et al, 2015) was not responsible for the later degradation of Cyclin B1 in the cytoplasm (Appendix Fig. S4C).

We conclude that the binding between the N-terminus of Cyclin B1 and the nucleosome acidic patch confers timely degradation on Cyclin B1.

## CRISPR/Cas9 gene editing to mutate the arginine anchor in Cyclin B1 required simultaneous mutation of p53

Chromosome binding is important for Cyclin B1 degradation to start as soon as the SAC is turned off; therefore, we sought to determine the effect on the fidelity of mitosis of eliminating chromosome binding by mutating the arginine anchor in endogenous Cyclin B1.

We used CRISPR/Cas9$^{D10A}$ to introduce the 4E7E mutation into both alleles of Cyclin B1 in RPE-1 Cyclin B1-mEmerald$^{+/+}$ cells (Fig. 7A). Our initial attempts did not result in any homozygous mutant clones (>2000 clones screened) but we did obtain clones when we co-transfected guide RNAs to mutate the tumour suppressor gene TP53 (Bowden et al, 2020; Chiang et al, 2016) (six potential homozygous clones from 132 colonies screened, as assessed by PCR). DNA sequencing confirmed homozygous 4E7E mutations in three independent clones (B1H3, 7A11, 7H10— hereafter referred to collectively as Cyclin B1$^{4E7E}$ clones, see source data). We also generated two RPE-1 Cyclin B1-mEmerald$^{+/+}$; TP53$^{-/-}$ cell lines as controls for the effect of knocking out p53 alone (9B2, 9B3—hereafter referred to collectively as p53$^{-/-}$ clones). For both Cyclin B1$^{4E7E}$ and p53$^{-/-}$ clones we assessed TP53 status by genomic sequencing (Appendix Fig. S5A,B), and tested for non-functional p53 by treating the cells with the p53-stabilising compound Nutlin-3A (Vassilev et al, 2004, Appendix Fig. S5C–F —see additional text for Appendix Fig. S5). Immunoblot analysis revealed that Cyclin B1 levels were considerably reduced in Cyclin B1$^{4E7E}$ clones (by ~ 95% with respect to parental cells, Appendix Fig. S5C, see below).

## The degradation of Cyclin B1$^{4E7E}$ is delayed and results in aneuploidy

We assayed Cyclin B1 localisation and degradation in RPE-1 Cyclin B1$^{4E7E}$-mEmerald$^{+/+}$ cells. In all Cyclin B1$^{4E7E}$ clones, line-profile analysis of confocal images of metaphase cells showed a strong reduction of Cyclin B1 localisation at chromosomes when

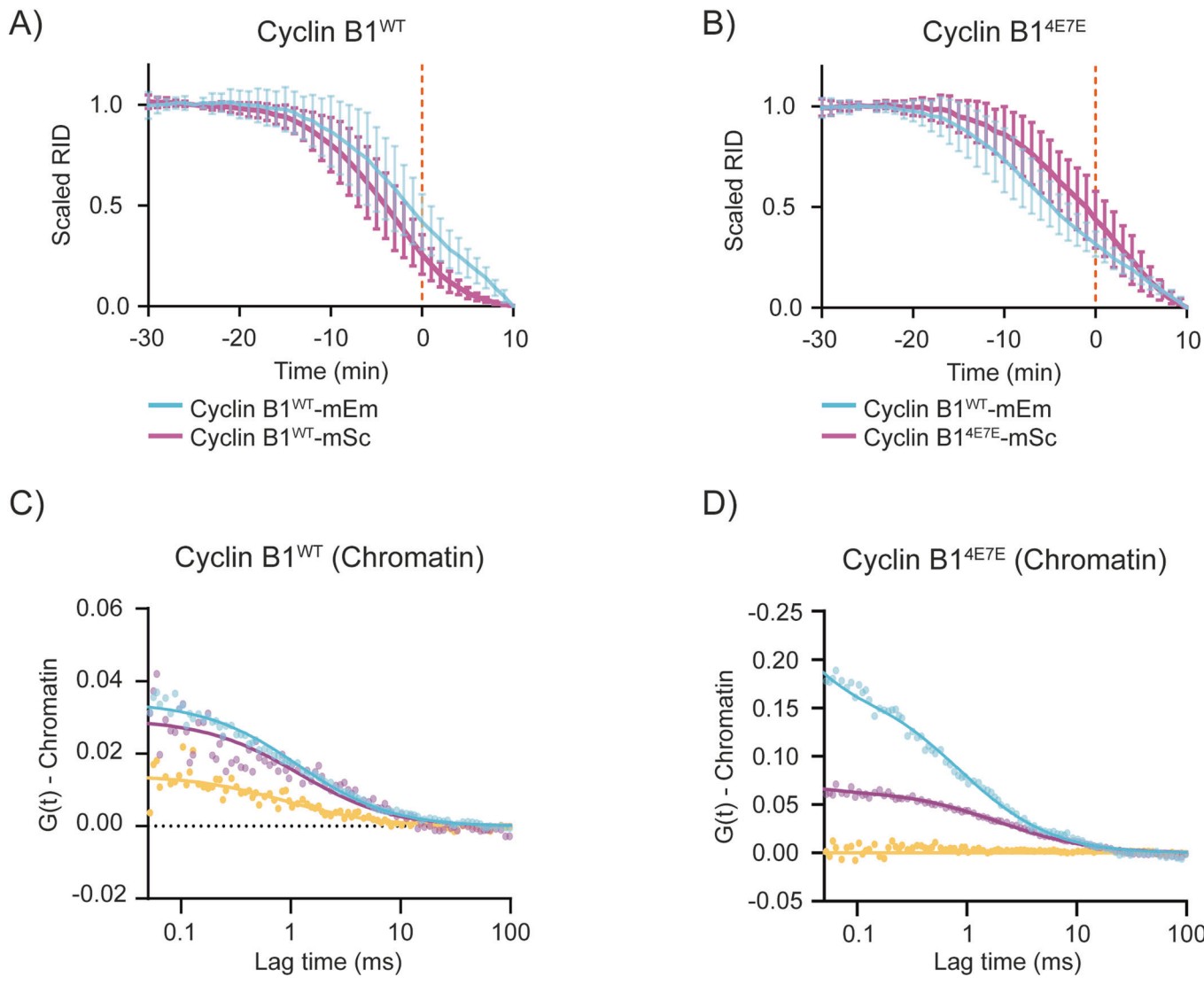

**Figure 5.  Cyclin B1 nucleosome localisation determines its timely degradation.**

(A, B). Plot of the fluorescence intensity of Cyclin B1 over time: $n \geq 17$ cells per condition, $N \geq 3$ independent experiments. In this and the following Cyclin B1 degradation graphs: Cyan -endogenous Cyclin B1-mEmerald; Magenta - ectopically expressed Cyclin B1-mScarlet, unless otherwise specified. Mean ± standard deviation are plotted. (C, D) Representative graphs of the autocorrelation of APC8-mScarlet (magenta) and ectopically expressed Cyclin B1-mEmerald (cyan) variant and the cross-correlation (yellow) between the two. Source data are available online for this figure.

compared to parental cell lines (Fig. 7B). Live-cell imaging revealed a delay of several minutes in Cyclin B1 degradation in Cyclin B1[4E7E] compared to parental Cyclin B1[WT], both when measuring overall levels of Cyclin B1 and at individual subcellular locations (Fig. 7C,D; Appendix Fig. S6A). In addition, Cyclin B1[4E7E] did not show any spatial pattern of differential degradation when comparing the chromosomes, spindle, centrosomes and cytoplasm (Fig. 7D). The delay in the degradation of Cyclin B1[4E7E] compared to Cyclin B1[WT] was maintained in cells treated with Reversine, indicating that the difference in timing of Cyclin B1[4E7E] degradation is not caused by a difference in SAC silencing (Appendix Fig. S6D). Note that the loss of p53 alone did not influence Cyclin B1 degradation (Appendix Fig. S6B). We obtained comparable results

with ectopic expression of Cyclin B1[4E7E]-mScarlet (Appendix Fig. S6C).

Quantifying total Cyclin B1 fluorescence in metaphase Cyclin B1[4E7E] clones, we measured a reduction of over 90% compared to the parental cell line, confirming our previous western blot result (Fig. 7E; Appendix Fig. S5C). This was likely to be caused by reduced transcription because there was a significant reduction in Cyclin B1 mRNA in the Cyclin B1[4E7E] cells compared to parental cell lines (Appendix Fig. S6E).

To gain further insight into the dynamics of Cyclin B1[4E7E] degradation, we compared its maximum degradation speed as well as the onset of degradation to those of the parental cell line (see Materials and Methods, Lu et al, 2015). Although the maximum

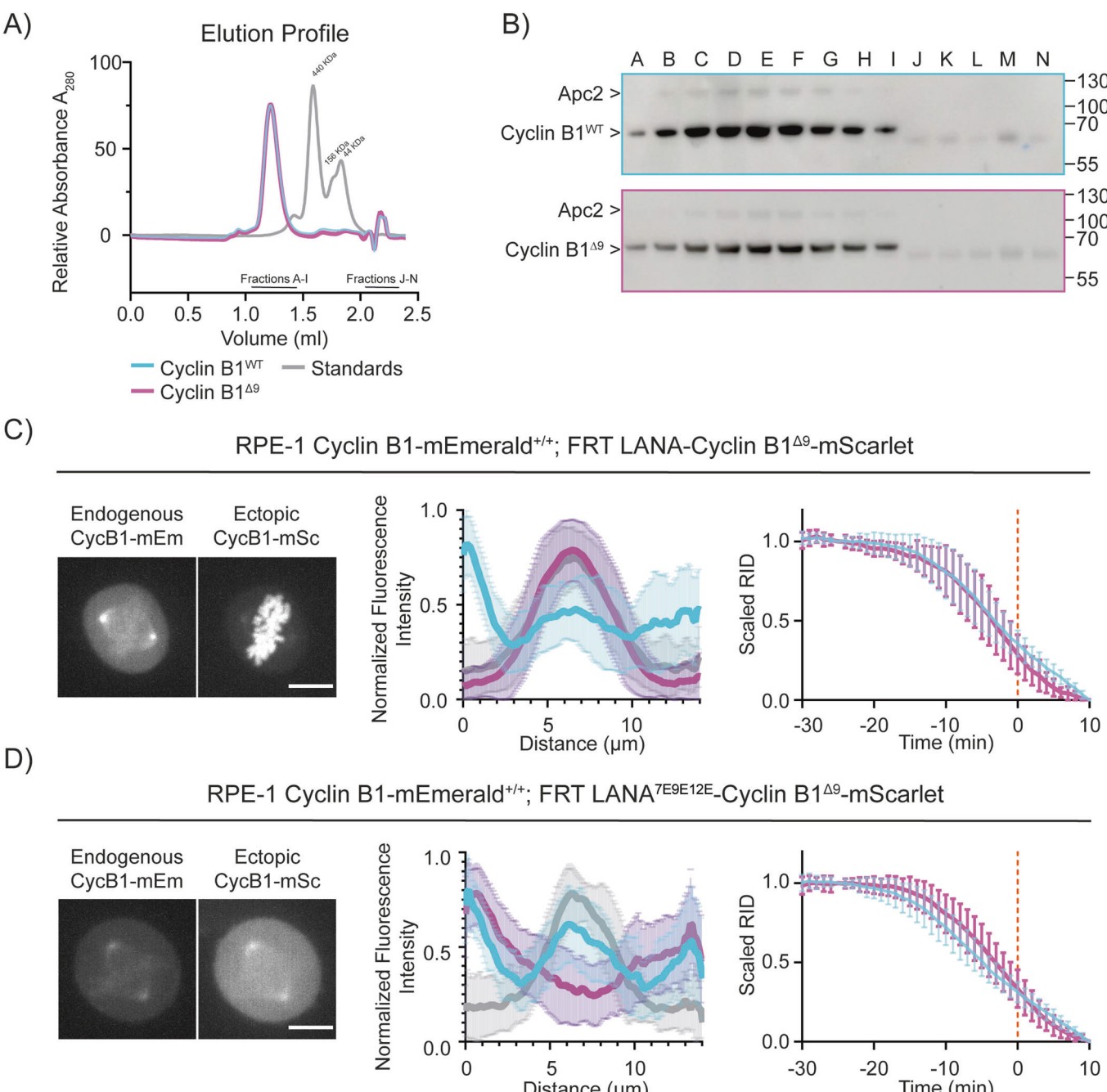

**Figure 6. Restoring Cyclin B1^Δ9 chromatin localisation rescues its degradation timing.**

(A) The elution profile of APC/C^CDC20 with Cyclin B1^WT (cyan) or Cyclin B1^Δ9 (magenta) on size-exclusion chromatography. Graph representative of $N = 3$ independent experiments. (B) Representative immunoblot of the size-exclusion chromatography is shown in (A). (C, D) Left: Maximum projections of confocal images representative of RPE-1 Cyclin B1-mEmerald^+/+ cells ectopically expressing the indicated variant of Cyclin B1-mScarlet. The scale bar corresponds to 10 μm. Middle: Graphs representing the pixel-by-pixel fluorescence intensity over a line going from centrosome to centrosome of RPE-1 Cyclin B1-mEmerald^+/+ (Cyan) cells ectopically expressing the indicated variant of Cyclin B1-mScarlet (Magenta). Grey indicates siR-DNA staining: $n \geq 38$ cells per condition, $N \geq 3$ independent experiments. Mean ± standard deviation are plotted. Right: Cyclin B1 degradation graph representing the fluorescence intensity of Cyclin B1 over time of RPE-1 Cyclin B1-mEmerald^+/+ (Cyan) cells ectopically expressing the indicated variant of Cyclin B1-mScarlet (Magenta): $n \geq 15$ cells per condition, $N \geq 3$ independent experiments. Mean ± standard deviation are plotted. Source data are available online for this figure.

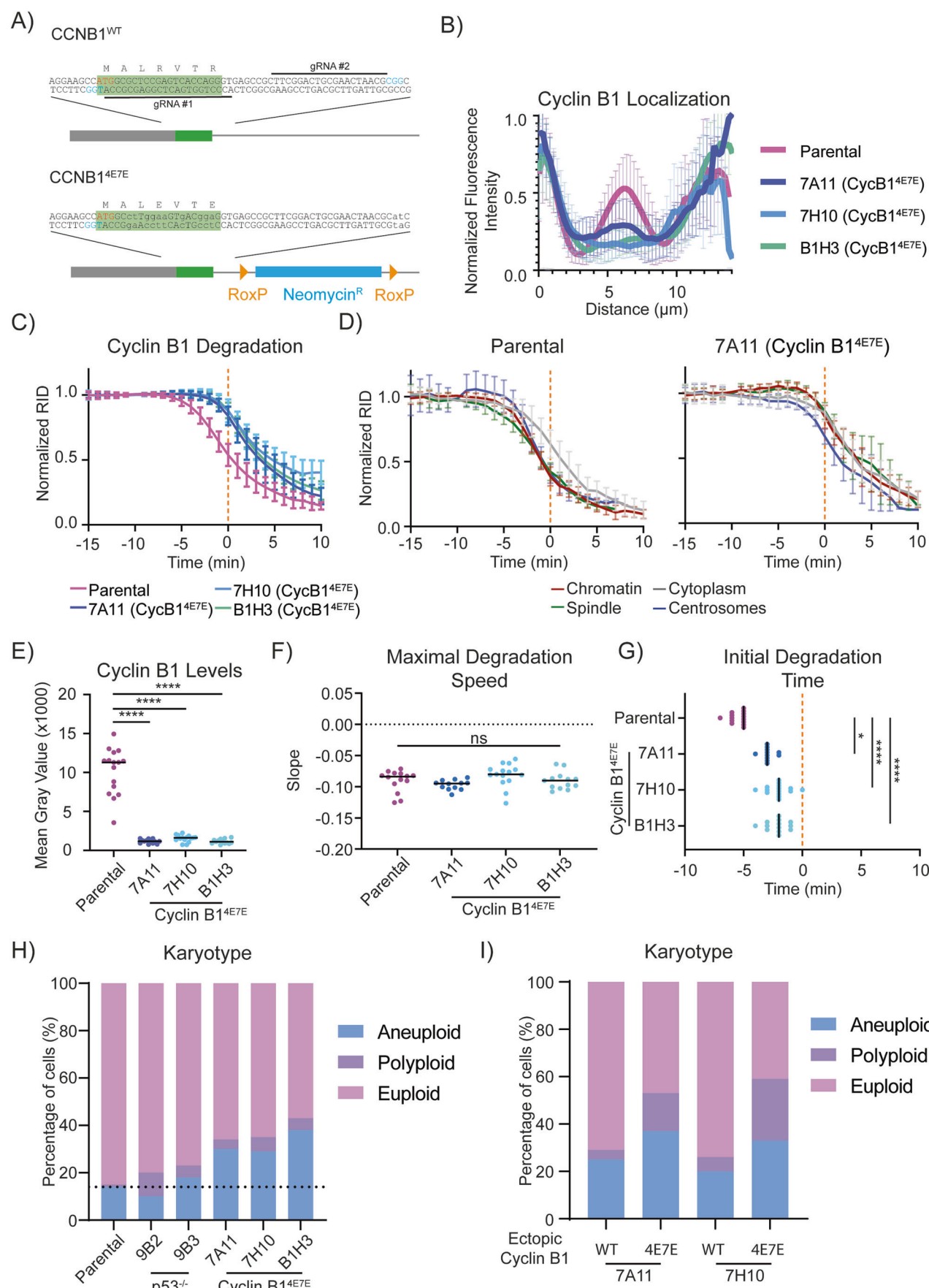

Figure 7. Endogenous Cyclin [4E7E] recapitulates ectopic Cyclin 4E7E and increases genomic instability.

(A) Schematic representation of the CRISPR strategy used to introduce the 4E7E mutation. The green box indicates CyclinB1's first exon, the blue box is a neomycin resistance cassette, grey box indicates the 5′UTR of CyclinB1. Yellow triangles indicate RoxP sites. The red text refers to CyclinB1's ATG, blue text refers to PAM sites. (B) Graphs representing the pixel-by-pixel fluorescence intensity over a line going from centrosome to centrosome of RPE-1 Cyclin B1-mEmerald[+/+] compared to Cyclin B1[4E7E] clones. For (panels B–G): $n \geq 12$ cells per condition, $N = 3$ independent experiments. Mean ± standard deviation are plotted. (C, D) Cyclin B1 degradation graph representing the fluorescence intensity of Cyclin B1 over time of RPE-1 Cyclin B1-mEmerald[+/+] compared to Cyclin B1[4E7E] clones. Mean ± standard deviation are plotted. (E) Dot plots representing the quantification of the mean fluorescence intensity of Cyclin B1 during metaphase in RPE-1 Cyclin B1-mEmerald[+/+] compared to Cyclin B1[4E7E] clones. (F, G) Dot plots representing the maximal degradation speed (E) or the initial degradation time (F) of RPE-1 Cyclin B1-mEmerald[+/+] compared to Cyclin B1[4E7E] clones. (H) Bar graph representing the karyotype classification from metaphase chromosome spreads of RPE-1 Cyclin B1-mEmerald[+/+] compared to p53[−/−] clones and Cyclin B1[4E7E] clones. The dotted line represents the level of aneuploidy in parental cells. $N = 3$ independent experiments. (I) Bar graph representing the karyotype classification from metaphase chromosome spreads of the indicated RPE-1 Cyclin B1[4E7E]-mEmerald[+/+] clones following 3 weeks of ectopic expression of either Cyclin B1[WT]-mScarlet or Cyclin B1[4E7E]-mScarlet. $N = 2$ independent experiments. Source data are available online for this figure.

degradation speed did not differ significantly between the Cyclin B1[4E7E] clones and the parental cell line, our analysis highlighted a significant difference in the timing of the onset of degradation (Fig. 7F,G; Appendix Fig. S6F,G).

Finally, we asked whether the Cyclin B1[4E7E] mutation had any effect on the fidelity of cell division. Analysis of chromosome spreads showed that 30 to 40% of the cells carrying the Cyclin B1[4E7E] mutation had become aneuploid compared to only 10 to 15% in the parental cells and p53[−/−] clones (Fig. 7H). To determine whether the increased aneuploidy originated from the arginine anchor mutation or the reduction in Cyclin B1 levels, we returned Cyclin B1 to levels comparable to those in normal cells by ectopic expression of Cyclin B1 from an FRT site in two independent clones of Cyclin B1[4E7E]. After 3 weeks of stable expression we found that Cyclin B1[4E7E] caused an increased in the number of polyploid cells, whereas the expression of wild-type Cyclin B1 had a considerably lower impact on the karyotype of Cyclin B1[4E7E] cells (Fig. 7I). This result demonstrates that it is the mutation of the arginine anchor rather than the reduction of Cyclin B1 levels that causes aneuploidy and polyploidy in RPE-1 cells.

## Discussion

The destruction of Cyclin B1 is the crucial step that initiates the events leading to chromosome separation and mitotic exit. Twenty years ago, we and others first showed that Cyclin B1 degradation is not homogeneous across the cell: in human cells, it disappears earlier from chromosomes, the spindle, and centrosomes than from the cytoplasm (Clute and Pines, 1999). Here, we have elucidated the mechanism behind this. We have found that the spatial pattern of Cyclin B1 degradation depends upon its binding to the acidic patch on nucleosomes through an N-terminal arginine anchor and this promotes binding to the APC/C and is necessary for its timely degradation in metaphase.

Combining CRISPR/Cas9 gene editing with FCCS has enabled us to measure dissociation constants in vivo. Thus, for the first time to our knowledge, we have measured the in vivo dissociation constant between Cyclin B1 and APC/C. This varies between ~80 nM at the chromosomes and ~180 nM in the cytoplasm (Fig. 1H; Appendix Table S1). Although these values are likely to be slightly lower than the actual dissociation constants due to the technical limitations of FCCS (Barbiero et al, 2022), they are close to the estimated 63 nM obtained in vitro using sea urchin Cyclin B1 and fission yeast APC/C (Carroll and Morgan, 2002).

Our data indicate that nucleosome binding is important for the APC/C to bind productively to Cyclin B1 as soon as the SAC is turned off. This is consistent with previous data in human cell lysates showing that APC/C ubiquitylation activity toward Cyclin B1 is enhanced in chromatin fractions compared to the cytoplasm (Sivakumar et al, 2014). We find that a Cyclin B1 mutant that is unable to bind nucleosomes still localises to the spindle and centrosomes but fails to engage the APC/C in a D-box-dependent manner and its degradation is delayed. Thus, an intriguing possibility is that binding to nucleosomes might expose or optimally present the D-box to the substrate binding site of the APC/C. Arginine 42 of the D-box is also important for chromosome localisation (Pfaff and King, 2013, our unpublished observation), which might indicate that D-box-mediated binding to nucleosome-bound APC/C enhances the avidity of Cyclin B1 interaction with chromatin.

Nucleosomes could enhance Cyclin B1 degradation by increasing the local concentration of Cyclin B1 and the APC/C. (Although APC3 and Cyclin B1 both bind the acidic patch, they are unlikely to compete in vivo given the abundance of nucleosomes (~6.23 μM) when compared to the much lower concentrations of both Cyclin B1 and APC/C (about 100 nM, Appendix Table S1, Beck et al, 2011). Increasing the local concentration of substrate and enzyme would both reduce the search space and increase the kinetics of ubiquitylation because partially ubiquitylated Cyclin B1 that dissociated from the APC/C would be prevented from diffusing away through binding adjacent nucleosomes and, in turn, be more likely to rebind the APC/C. Congruent with this idea, we observed that recruiting securin to chromatin is sufficient to enhance its degradation, and changing the topology of the interaction between Cyclin B1 and nucleosomes (by moving the LANA peptide from N- to C- terminus) has no evident effect on Cyclin B1 degradation.

The APC3 loop that we have determined binds to the nucleosome acidic patch is implicated in APC/C activation. This loop contains phosphorylation sites that are crucial for CDC20 recruitment and APC/C activity in vitro (Zhang et al, 2016); therefore, one mechanism by which nucleosome binding could enhance substrate recognition would be if this bypasses the need for APC3 phosphorylation. In this respect, it is intriguing that previous work from the Gorbsky lab found chromatin-bound APC/C is more efficient at ubiquitylating Cyclin B1 despite being hypophosphorylated (Sivakumar et al, 2014). Furthermore, it is possible that the binding between APC/C and nucleosomes may be regulated. Serine 379 is within the APC3 loop arginine anchor motif and phosphorylation here could interfere with binding to the acidic

patch. Serine 379 is phosphorylated by AMPKα2 (Banko et al, 2011), which means it could respond to energy levels in the cell, but it also conforms to an Aurora kinase consensus site and so could be regulated in mitosis.

A third possibility is that nucleosomes neutralise the acidic charge on DNA. This would provide an environment that favours substrate recognition by APC/C^CDC20 because polyanions have been shown to promote the dissociation of CDC20 from the APC/C in vitro (Mizrak and Morgan, 2019).

We do not know whether nucleosome binding is a conserved mechanism to mediate APC/C binding to Cyclin B1. The N-terminal sequence of Cyclin B1 diverges in non-vertebrates and plants but it is generally arginine-rich, which could mean that nucleosome binding is also conserved. One exception is Drosophila, where the N-terminal arginines of Cyclin B1 are not conserved and it is notable that Drosophila Cyclin B first disappears on the spindle poles and only later on chromosomes (Huang and Raff, 1999). Although we did not directly investigate the role of spindle and centrosome localisation of Cyclin B1 in its degradation, we found that preventing Cyclin B1 binding to chromatin delays its degradation throughout the cell, arguing that chromatin is the major site of Cyclin B1-APC/C binding in human cells. This raises the question of why Cyclin B1 also appears to be degraded faster at the centrosomes and spindle. A possible explanation is the fast diffusion of Cyclin B1 between centrosomes, spindle and chromosomes, such that Cyclin B1 could be rapidly ubiquitylated by a chromosome-bound pool of APC/C and then degraded anywhere between the chromatin and the spindle apparatus. Our FCCS data, however, show that Cyclin B1–APC/C complexes are present at centrosomes and spindles. Thus, we favour the model that nucleosomes promote the formation of a Cyclin B1–APC/C complex that can diffuse to other subcellular locations.

Although an arginine anchor is not present in other cyclins, nor in securin and NEK2A, some APC/C substrates such as KIFC1 (Kinesin-like protein C1), BARD1 (BRCA1-associated RING domain protein 1), TPX2 (Targeting protein for Xklp2), and ZC3HC1 (Zinc finger C3HC-type protein 1), are known to bind the nucleosome acidic patch (Singh et al, 2014; Skrajna et al, 2020; Song and Rape, 2010; Stewart and Fang, 2005; von Klitzing et al, 2011), and thus could share an APC/C interaction mechanism similar to that of Cyclin B1. The APC/C also has an important role in G1 phase (Wirth et al, 2004) and in neurodifferentiation, where nucleosome-binding could be important to target chromatin-associated proteins such as Ki67 (Antigen Kiel 67), Top2a (DNA topoisomerase IIα) and the chromosomal passenger complex that accumulate in APC/C mutant mice and G1 phase of CDH1-mutant RPE-1 cells (Ledvin et al, 2023). Multiple mechanisms for chromatin recruitment of the APC/C may exist, however, since Oh and colleagues reported that APC/C binding to promoters of ES cells depends on WDR5 to regulate G1 transcription in human embryonic stem cells (Oh et al, 2020).

We found that mutating the arginine anchor of Cyclin B1 results in aneuploidy in otherwise genomically stable RPE-1 cells. We excluded the alternative explanation that this effect is due to lower levels of Cyclin B1, in agreement with previous observations that mitosis is not perturbed in non-transformed somatic cells with very low levels of Cyclin B1 (Bellanger et al, 2007; Hégarat et al, 2020). It is likely that the reduction in Cyclin B1^4E/7E levels is a compensatory mechanism. In early mitosis, securin and Cyclin B1 prevent the premature segregation of sister chromatids by directly binding and inhibiting separase, and Cyclin B1-Cdk1 inhibits cytokinesis. At the end of metaphase, separase must quickly cleave the cohesin ring that holds chromatids together to ensure chromosome segregation is coordinated with mitotic exit and cytokinesis. Delaying Cyclin B1 degradation would prolong separase inhibition by Cyclin B1, which would result in chromosome segregation defects and consequent aneuploidy. Higher than normal Cyclin B1 levels in anaphase would also perturb the formation of the cytokinetic furrow and could lead to cytokinesis failure and consequent polyploidy. Thus, lowering the overall amount of Cyclin B1 in cells would guarantee low Cdk1 activity during mitotic exit despite delayed degradation of Cyclin B1

In summary, we have identified the mechanism behind the spatial control of Cyclin B1 destruction by the APC/C, and our results indicate that this is important to ensure genomic stability. From a broader perspective, one can envisage how localised proteolysis can help to generate gradients of enzyme activity that are particularly important in a mitotic cell where many of the intracellular barriers to diffusion have been removed.

## Methods

### Cell culture and drug treatment

hTERT RPE-1 FRT/TO cells were cultured in F12/DMEM medium (Sigma-Aldrich) supplemented with GlutaMAX (Invitrogen), 10% FBS (Gibco), 0.35% sodium bicarbonate, penicillin (100 U/ml), streptomycin (100 μg/ml) and Fungizone (0.5 μl/ml). Cells were maintained in a humidified incubator at 37 °C and 5% CO$_2$ concentration. For live-cell imaging experiments cells were imaged in Leibovitz L-15 medium (Thermo Fisher) supplemented with 10% FBS, penicillin (100 U/ml) and streptomycin (100 μg/ml).

In the indicated experiments, cells were stained with 20 nM siR-DNA (Spirochrome) following the manufacturer's protocol or treated with the following drug concentrations: 100 nM paclitaxel (Sigma-Aldrich); 10 μM MG132 (Selleckchem); 12 μM proTAME (Sigma) and 200 μM APCin (Sigma); 5 μM Reversine (Selleckchem); 50 nM Cenp-E inhibitor (GSK923295 - Selleckchem); 10 μM Nutlin-3A (Sigma). Gene expression was induced using 1 μg/ml tetracycline (Calbiochem). In FCCS experiments tetracycline was added 3 h before imaging, in all other experiments tetracycline was added 16 h before imaging. Cells were exposed to Nutlin-3A for 24 h prior to the experiment. MG132, GSK923295, APCin, and proTAME were added 0.5–1 h before the experiment began. Reversine and paclitaxel were added immediately before the experiment.

### Gene editing

To tag APC8, one million RPE-1 FRT/TO cells or RPE-1 Cyclin B1-mEmerald^+/+ FRT/TO cells were transfected using 500 ng of a modified version of the PX466 'All-in-One' plasmid containing Cas9^D10A-T2A-mEmerald and gRNAs targeting APC8 (5′-CCA-CACGCAGAGTTTCTCCA-3′ and 5′-GTCTTCTGTCACGCCA-TAGT-3′). The all-in-one plasmid was co-transfected with 500 ng of repair plasmid encoding G-S-A-G-S-A-mScarlet flanked by two 500 bp arms, homologous to the genomic region around the Cas9

cutting site. All CRISPR repair templates were cloned into the pUC57-Kan plasmid. Seventy-two hours post transfection, 50,000 mEmerald positive cells were sorted in a 1 cm well and expanded for 1 week before a second sorting of single cells in 96-well plates. The mScarlet tag was identified through PCR using forward 5′-AAGTGGAAAGCCTACCTTGG-3′ and reverse 5′-GCTGGCTTG AGAGTAGCCAAC-3′ primers. PCR products of positive clones were sequenced using the same primers.

To generate Cyclin B1$^{4E7E}$ and p53$^{-/-}$ cell lines, RPE-1 FRT/TO cells or RPE-1 Cyclin B1-mEmerald$^{+/+}$ FRT/TO cells were transfected using Cas9$^{D10A}$ RiboNucleoProteins (RNPs—Integrated DNA technology). Briefly, sgRNA was assembled by mixing equal volumes of individual gRNA 100 μM with tracr-RNA 100 μM. For Cyclin B1 the following gRNAs were used: 5′-CCTGGTGACTCG-GAGCGCCA-3′ and 5′- CTTCGGACTGCGAACTAACG-3′. For TP53 we used 5′-TCCACTCGGATAAGATGCTG-3′ and 5′-AAATTTGCGTGTGGAGTATT-3′ (Chiang et al, 2016). The mixture was incubated at 95 °C for 5 min, and then kept at RT for 40 min to anneal sgRNAs. About 4 uL of annealed sgRNA were mixed with 2.5 μL of 62 nM Cas9$^{D10A}$ protein to obtain RNPs. Two million cells were transfected with 18.8 pmol of each RNPs (two targeting CyclinB1, two targeting TP53), together with 6 μg of a repair plasmid (in the case of Cyclin B1$^{4E7E}$ clones) designed with a wobbled first exon of Cyclin B1 with the 4E7E mutation and a Neomycin resistance cassette, flanked by two 500 bp arms homologous to the genomic region around the Cas9 cutting site. Seventy-two hours post transfection, Cyclin B1$^{4E7E}$ cells were selected for 72 h with Nutlin-3A 5 μM and for 7 days with 0.4 mg/ml Geneticin (Gibco). p53$^{-/-}$ cells were not transfected with any repair template and did not receive the Geneticin selection. Single cells were then sorted in 96-well plates. The presence of the 4E7E mutation was identified through PCR and sequencing using forward 5′- GCCTTTCATGAACTATATTATTGC-3′ and reverse 5′-AGCCGCCGCATTGCATCAGC-3′ primers. TP53 status was assessed by PCR and sequencing using forward 5′-CTAGTGGGGTTGCAGGAGGTG-3′ and reverse 5′- TAAGCAG-CAGGAGAAAGCCC-3′ primers. To distinguish the two alleles of TP53, PCR products were cloned into the TOPO2.1 plasmid prior to sequencing, according to the manufacture's protocol.

For ectopic expression of Cyclin B1-mScarlet, Securin-mScarlet, and Cyclin B1-mEmerald variants, RPE-1 FRT/TO CyclinB1-mEmerald$^{+/+}$, RPE-1 FRT/TO CyclinB1-mEmerald$^{+/+}$; APC8-mScarlet$^{+/+}$ or RPE-1 FRT/TO APC8-mScarlet$^{+/+}$ cell lines were co-transfected with the relevant cDNA cloned into pcDNA5-FRT/TO (p1795) and pOG44 (Invitrogen) using a 1:5 ratio (using 1 μg total DNA per million cells). All transfections were followed by a 2-week selection using Geneticin (Gibco) 0.4 mg/ml. Cells expressing Cyclin B1-mScarlet variants were sorted and only low-expressing cells were selected for further analysis. Cells expressing the Cyclin B1$^{R42AL45A}$-mEmerald mutant were not sorted due to the cytoxicity of the construct. For ectopic expression of Cyclin B1-mScarlet variants in Cyclin B1$^{4E7E}$ clones, RPE-1 FRT/TO Cyclin B1$^{4E7E}$-mEmerald$^{+/+}$ (clones 7A11 and 7H10), were co-transfected with the relevant cDNA cloned into a modified version of pcDNA5-FRT/TO where neomycin resistance was swapped with mTurquoise2 and pOG44 (Invitrogen) using a 1:5 ratio (using 1 μg total DNA per million cells). Cells expressing Cyclin B1-mScarlet variants were sorted using the mTurquoise2 signal. In Cyclin B1 overexpression experiments, cells were exposed to tetracycline for 2 weeks, sorted for mScarlet

expression and grown an additional week in tetracycline before performing chromosome spreads (see below).

All transfections were performed by electroporation using a Neon Transfection System (Invitrogen) with two pulses at 1400 V for 20 ms.

## Protein extraction and immunodepletion

Adherent cells were treated with trypsin and incubated in lysis buffer (150 mM NaCl, 50 mM Tris pH 7.4, 0.5% NP-40) supplemented with HALT protease/phosphatase inhibitor cocktail (Thermo Fisher Scientific) for 30 min at 4 °C before clarification. Lysates were clarified by centrifugation (14,000 × $g$, 20 min, 4 °C) and quantified using Bradford Reagent (Bio-Rad Laboratories) according to the manufacturer's instructions.

For APC4 immunodepletion, 200 μg of clarified lysates were diluted to a final concentration of 2 μg/μl and incubated with 30 μl of Dynabeads prebound to 3.75 μg of either anti-APC4 antibody (Movarian Biotech) or Mouse IgG, in a total volume of 100 μl for 2 h at 4 °C. 40 μg of either the immunodepleted lysate or the input was used for SDS-PAGE.

## Immunoblotting

Cell lysates were separated through SDS-PAGE on a 4–12% NuPAGE gel (Invitrogen) and transferred to an Immobilon-FL polyvinylidene fluoride membrane (IPFL00010, Millipore). The membrane was blocked with 5% milk, 0.1% Tween in PBS and incubated overnight with primary antibodies at 4 °C in 2.5% milk and 0.1% Tween in PBS. The following day, the membrane was washed with 0.1% Tween in PBS and incubated with secondary antibodies in 2.5% milk and 0.1% Tween in PBS for 1 h at RT. Membranes were visualised with a LI-COR Odyssey CLx scanner (LI-COR Biosciences). Immunoblot quantification in Fig. EV1F was performed by calculating the area under the curve using Fiji's 'Gel' plugin. Values were adjusted by fitting to a straight-line function obtained by immunoblotting serial dilutions of protein lysate.

In size-exclusion chromatography, membranes were blocked overnight in 5% BSA. The immunoblots were developed using the ImageQuant 800 (Amersham).

Primary antibodies were used at the indicated concentrations: anti-APC2 (1:1000, Movarian Biotech), anti-APC4 (1:1000, Movarian Biotech), anti-APC8 (1:1000, D5O2D – Cell Signalling), Strep-MAB (1:10000, IBA 2-1507-001 - IBA), anti-p53 (1:1000, DO-1 – Santa Cruz), anti-p21 (1:1000, 12D1 – Cell Signalling), anti-β Tubulin (1:10000, Ab6046 - Abcam), anti-Cyclin B1 (1:1000, GNS1 - Santa Cruz).

Secondary antibodies were: IRDye800CW donkey anti-mouse (926-32212, LI-COR), IRDye800CW donkey anti-rabbit (926-32213, LI-COR), IRDye680CW donkey anti-mouse (926-68072, LI-COR), and IRDye680CW donkey anti-rabbit (926–68073, LI-COR), HRP-conjugated goat anti-mouse (Amersham), HRP-conjugated goat anti-rabbit (Amersham). Secondary antibodies were all used at 1:10,000.

## qPCR

RNA was extracted from exponentially growing cells using Monarch Total RNA extraction kit (New England Biolabs),

following the manufacturer's protocol. About 3 μg of RNA were retrotranscribed to cDNA using Superscript III Reverse Transcriptase (Invitrogen) according to the manufacturer's protocol. SYBR Green qPCR on cDNA samples was performed by Syd Labs using the following primers: GADPH Fw: 5′- ACAACTTTGGTATCGTG GAAGG-3′; GADPH Rv: 5′-GCCATCACGCCACAGTTTC-3′; Cyclin B1 Fw: 5′- AACAAGTATGCCACATCGAAGC-3′; Cyclin B1 Rv: 5′-TACACCTTTGCCACAGCCTT-3′.

## Expression of proteins in insect cells

Codon-optimised cDNAs of wild-type and mutant Cyclin B1-StrepII and N-terminal domain GST-Cyclin B1-His in pFastBac1 vectors were ordered from GeneArt (Thermo Fischer Scientific) and transformed into MultiBac cells for insect cell expression. Sf9 cells were used to create the baculoviruses which were then used to infect High Five cells at a cell density of $1.5 \times 10^6$ cells/mL. High Five cells were incubated for 72 h at 27 °C, 130 rpm.

To express the APC/C, three separate baculoviruses were used to infect High Five cells: the first contained APC/C subunits APC5, APC8, APC10, APC13, APC15, APC2 and APC4-strepII, (gift of David Barford). The second contained APC11 and phosphomimic mutant APC1: Ser364Glu, Ser372Glu, Ser373Glu and Ser377Glu (Zhang et al, 2016). The final baculovirus contained subunits APC3, APC6, APC7, APC12 and APC16. Cells at a density of $2 \times 10^6$ were infected with these baculoviruses and incubated for 72 h at 27 °C, 130 rpm.

To express the APC3 loop, a codon-optimised cDNA of wild-type APC3 loop[WT] (APC3 residues 177–446) was ordered in a pFastBac1 vector from GenScript with an N-terminal strepII tag. The mutant and spy-tagged forms of the APC3 loop were created via cloning, the APC3 loop[Spy-tag] contain a 27 residue GSA linker between the spytag and the APC3 loop construct. All three constructs were used to create the baculoviruses to infect High Five cells at a cell density of $1.5 \times 10^6$ cells/mL High Five cells were incubated for 72 h at 27 °C, 130 rpm.

## Protein purification

All purification steps were carried out at 4 °C.

### APC/C

Cell pellets were thawed on ice in wash buffer (50 mM HEPES pH 8.3, 250 mM NaCl, 5% glycerol, 2 mM DTT, 1 mM EDTA and 2 mM Benzamidine supplemented with 0.1 mM PMSF, 5 units/mL Benzonase and an EDTA-free protease inhibitor (Roche)). After sonication, the cells were centrifuged for 1 h at $48,000 \times g$. The supernatant was bound to a $3 \times 5$ mL StrepTactin Superflow Plus cartridge (Qiagen) using a 1 mL/min flow rate. The column was washed extensively with APC/C wash buffer and eluted with a wash buffer containing 2.5 mM desthiobiotin (IBA-Lifesciences). Fractions containing APC/C were incubated with tobacco etch virus (TEV) protease overnight. The sample was prepared for anion-exchange chromatography by a two-fold dilution with saltless Buffer A (20 mM HEPES pH 8.0, 125 mM NaCl, 5% glycerol, 2 mM DTT, 1 mm EDTA). The sample was then loaded onto a 6 mL ResourceQ anion-exchange column (GE Healthcare), and the column was washed extensively with buffer A. An elution gradient with Buffer B (20 mM HEPES pH 8.0, 1 M NaCl, 5% glycerol, 2 mM DTT, 1 mM EDTA) was used to elute the APC/C. The APC/C was

concentrated and centrifuged (Optima TLX Ultracentrifuge) at 40,000 rpm for 30 min.

### APC3 loop

Wild-type, mutant and Spy-tagged APC3 loop cell pellets were thawed in wash buffer (50 mM HEPES pH 7.5, 200 mM NaCl, 5% glycerol and 0.5 mM TCEP supplemented with 5 Units/mL Benzonase and an EDTA-free protease inhibitor cocktail (Roche)). Cells were sonicated and centrifuged at $41,000 \times g$ for 1 h. The supernatant was loaded onto a 5 mL StrepTactin Superflow Plus cartridge (Qiagen) using a 1 mL/min flow rate, washed with wash buffer, and eluted with wash buffer containing 2.5 mM desthiobiotin. The APC3 loop was concentrated and cleaved overnight with tev at 4 °C. The samples were then loaded onto a HiLoad 16/600 Superdex 75 pg column (Cytiva) in a gel filtration buffer (20 mM HEPES pH 7.5, 150 mM NaCl, 0.5 mM TCEP).

### CDC20

CDC20 was purified as previously described (Zhang et al, 2016).

### Cyclin B1

Wild-type and mutant Cyclin B1 cell pellets were thawed in Cyclin B1 wash buffer (50 mM HEPES pH 8.0, 500 mM NaCl, 5% glycerol and 0.5 mM TCEP supplemented with 5 Units/mL Benzonase and an EDTA-free protease inhibitor cocktail (Roche)). Cells were sonicated and centrifuged at $55,000 \times g$ for 1 h. The supernatant was loaded onto a 5 mL StrepTactin Superflow Plus cartridge (Qiagen) using a 1 mL/min flow rate and washed with Cyclin B1 wash buffer and eluted with wash buffer containing 2 mM desthiobiotin. Cyclin B1 was concentrated and loaded onto a HiLoad 16/600 Superdex 200 pg column (Cytiva).

### Cyclin B1[NTD]

This construct contains the first 95 amino acids of Cyclin B1 with an n-terminal GST-tag and a c-terminal His-tag. The cells were lysed in wash buffer (50 mM HEPEs pH 8.0, 200 mM NaCl, 20 mM imidazole, 0.5 mM TCEP, 5% glycerol) supplemented with an EDTA-free protease inhibitor cocktail (Roche) and 5 U/mL Benzonase. The cells were sonicated for 10 min using a 3 s on, 5 s off pulse and centrifuged for 1 h at $55,000 \times g$, this was all carried out at 4 °C. The filtered supernatant was loaded onto a 5 mL HisTrap HP column (Cytiva), and eluted using a gradient of 20 mM imidazole to 300 mM over 150 mL. The construct was cleaved with a 50:1 ratio of Cyclin B1[NTD]:HRV-3C overnight at 4 °C. It was then loaded onto a 5 mL GST column (Cytiva) using a 0.3 mL/min flow rate. After this, the supernatant was loaded onto a Superose 6 increase 10/300 GL column in gel filtration buffer (20 mM HEPEs pH 8.0, 150 mM NaCl, 0.5 mM TCEP).

### H2A-H2B-SpyCatcher

The H2A-H2B-SpyCatcher fusion cell pellet was thawed in wash buffer (50 mM HEPEs pH 8.0, 2 M NaCl, 5% glycerol, 0.5 mM TCEP). Cells were sonicated to lyse and centrifuged as previously mentioned. They were loaded onto a 5 mL StrepTactin Superflow Plus cartridge (Qiagen), washed with wash buffer and eluted with wash buffer supplemented with 3 mM desthiobiotin. The eluted fractions were injected onto a Superdex s75 increase 16/600 column in gel filtration buffer (20 mM HEPEs pH 8.0, 2 M NaCl, 0.5 mM TCEP).

## Peptide preparation

A peptide of the first 21 residues of human Cyclin B1, MAL-RVTRNSKINAENKAKINM, (NeoBiotech) was dissolved in water.

The first 23 residues of Kaposi's sarcoma-associated herpesvirus, latency associated nuclear antigen (LANA) protein, MAPPGMRLRSGRSTGAPLTRGSC (GenScript) was dissolved in water.

## NCP assembly

For NCP147, recombinant H2A, H2B, H3 and H4 were expressed separately in *Escherichia coli*, all histones were from *Homo sapiens* apart for H2B which was from *Xenopus laevis*. The histones were purified from inclusion bodies and assembled with Widom 601 147 bp DNA sequence as previously described (Luger et al, 1999). For the Spycatcher nucleosome used in the structural determination of the NCP:APC3 loop structure the purified H2A-H2B-SpyCatcher fusion construct was purified as stated then refolded with H3H4 tetramers as described, (Luger et al, 1999), at a 1:2.2 ratio of H3H4 tetramer:H2A-H2B-SpyCatcher. The octamer obtained was run through a Superdex s200 increase 16/600 column in the same gel filtration buffer. All four histone sequences were from *Homo sapiens*. DNA wrapping was carried out as previously mentioned (Luger et al, 1999). The DNA used for the SpyCatcher NCP was the Widom 601 147 bp sequence flanked on either side by 32 bp of DNA.

## On column crosslinking of the NCP and APC3 loop

To form NCP:APC3 loop complex, the SpyTag:SpyCatcher system (Keeble et al, 2017) was employed. First, a titration of the SpyTag APC3 loop was carried out with the SpyCatcher NCP to find the ratio at which the SpyCatcher is saturated with covalently bound SpyTag with no or minimal excess. The samples were run in a 4–12% Bis-Tris gel using 1X MES running buffer at 180 V. A 1:1 volume ratio of NCP to APC3 loop was taken further and scaled up to 200 µL sample volume containing 1 µM NCP and the equivalent volume of APC3 loop, made up to 200 µL with 20 mM HEPEs pH 7.5, 50 mM NaCl, 0.5 mM TCEP. About 900 µL of 0.3% glutaraldehyde was injected into a Superose 6 increase 10/300 GL column equilibrate in 20 mM HEPEs, 50 mM NaCl, 0.5 mM TCEP. This was run at a 0.12 mL/min flow rate until 3 mL, then the protein sample was injected onto the column and run using the same flow rate. The fractions were quenched with 50 mM Tris-HCl pH 8.0. The desired fractions were pooled and concentrated to 0.92 µM.

## Cryo-EM grid preparation

Quantifoil R1.2/1,3 Cu 300 grids were glow discharged using 15 mA for 1 min on each side of the grid using an Easiglow (Pelco). For the NCP[CbNT] structure, 1.68 µM of NCP was mixed with 411 µM of CyclinB1[(1-21)] peptide in 20 mM HEPES pH 7.5, 20 mM NaCl, 0.1 mM TCEP. For the NCP-APC3 loop structure 0.92 µM of crosslinked NCP-APC3 loop sample was used in 20 mM HEPEs pH 8.0, 50 mM NaCl, 0.5 mM TCEP, 200 µM of cryoprotectant peptide developed by us was mixed with the sample prior to application to the grid [MALKVTKNSKINAENKAKINM]. For both samples 2 µL of the sample was added to the grid. After a 5 s wait the grids were blotted for 5 s with a blot force of 3, at 4 °C, 100% humidity. The grids were frozen in liquid ethane using a Vitrobot Mark IV (Thermo Fisher).

## Cryo-EM data collection and processing

We collected 4589 EER images of the NCP[CbNT] sample on our in-house Glacios Cryo-TEM 200 kV equipped with a Falcon4i detector operated in counting mode at a pixel size of 0.567 Å per pixel. Movie stacks were frame-aligned and binned four times. Images with a resolution better than 6 Å and a total motion of 30 pixels (estimated during frame alignment) were selected for further processing. A blob picker was used for initial particle picking during the live processing, and for obtaining initial 2D class averages. Templates generated from the latter, were used for template picking and TOPAZ (Bepler et al, 2019; Punjani et al, 2017) training and picking with cryoSPARC. Selected particles from 2D classifications performed with particles picked with the different methods explained above were pooled together and duplicated particles were removed by using the 'remove duplicates' function implemented in cryoSPARC. This step removed doubly picked particles within 100 Å (the length of the NCP is ~100 Å). Particles were piped into the 'ab initio model' function implemented in cryoSPARC and then imported in RELION-4 (Zivanov et al, 2022) by using pyem and the csparc2star.py script by Daniel Asarnow: David Asarnow, Eugene Palovcak and Yifan Cheng (2019). asarnow/pyem: UCSF pyem v0.5 (v0.5). Zenodo. https://doi.org/10.5281/zenodo.3576630) for 3D refinements. Map improvement was performed by performing sequential 3D classification without alignment (T = 4, T = 40 and T = 80) with a soft mask coupled with 3D refinements, Bayesian polishing steps and Ctf refinements in RELION. This yielded a map at the resolution of 2.5 Å (Fig. S5).

The PDB-ID:3LZ0 was used as initial template for 25 building in Coot (Casañal et al, 2020). Cyclin B N-terminus was built de novo based on the excellent quality of our cryo-EM map. Structural model refinement was performed with PHENIX real-space refinement (Afonine et al, 2018) at the resolution of 2.5 Å.

For the NCP-APC3 loop[375-381] sample, we collected 64,319 mrc images on the Francis Crick Institute LonCEM Krios Cryo-TEM 300 kV equipped with a K3 counting detector using a pixel size of 0.52 Å per pixel. Movies were motion corrected and binned two times, images with resolution better than 5 Å with a max in frame motion less than five were selected, a final 35,852 images were used for further processing. Live processing was carried out using CryoSPARC live, particles were picked with templates generated from a different nucleosome collection. Selections from the initial 2D classes were used to train Topaz (Bepler et al, 2019; Punjani et al, 2017) and the trained algorithm was used to repick from the micrographs in cryoSPARC. Duplicate particles were removed, and then the resulting particles were used to do an initial 'ab initio' with three classes. The best class was chosen, and the resulting 2D classes were exported from cryoSPARC into RELION (Zivanov et al, 2022) in the same way as mentioned previously. Once in RELION we carried out two cycles of re-extraction from the micrographs, using a box size of 100 and binned four times. This

was followed by 3D refinement, followed by 3D classification with three classes with T = 4. We selected the class which had the best peptide density but also the best density for the nucleosome DNA ends. We re-extracted again, binning two times followed by 3D refinement. We then carried out two cycles of: Bayesian polishing, 3D refinement, CTF refinement and 3D refinement. We used a box size of 216 and binned two times during the polishing. We then masked around the area with the APC3 loop peptide bound and ran a particle signal subtraction with particle re-centring and classification into three classes. We selected the class with the best density for the APC3 loop peptide. This gave us a map of 2.55 Å, we carried out postprocessing with a B-factor of -50 for model building. To build the model, we docked in the previous map (NCP$^{CycB}$) as an initial template in Coot (Casañal et al, 2020). The APC3 loop was built de novo and structural model refinement was carried out with PHENIX real-space refinement (Afonine et al, 2018) using a resolution of 2.6 Å.

## Size-exclusion chromatography

Samples were run through a Superose 6 Increase 3.2/300 2.4 mL column on an ÄKTAmicro system in 20 mM HEPES pH 8.0, 150 mM NaCl, 0.5 mM TCEP in a 30 µL reaction volume. About 1.5 µM APC/C, 3.3 µM CDC20 and 3.3 µM of Cyclin B1 were mixed and placed on ice for 5 min before injection. About 18 µM was injected when running the Cyclin B1 constructs alone. Selected fractions were run on SDS-PAGE and then transferred to a PVDF membrane.

## Electrophoretic mobility shift assay

For EMSAs of the NCP with Cyclin B1, 0.5 µM of NCP147 was mixed with full-length Cyclin B1 or the N-terminal mutants at a ratio of 1:0, 1:2, 1:4, 1:8 and 1:16 in a 6 µL reaction volume. The reaction buffer used was 20 mM HEPES pH 7.5, 50 mM NaCl and 0.1 mM TCEP. Reactions were incubated on ice for 10 min before adding 5% (v/v) sucrose. The samples were run on a 5% polyacrylamide gel in 0.25X TBE running buffer at 100 V for 90 min. The gels were then stained with SYBR safe and scanned using a Typhoon FLA 9500 imager. Quantification was done using ImageJ and the band intensities were normalised compared to the band intensity of the NCP without protein complex.

For EMSAs with the APC/C-CDC20-Cyclin B1$^{NTD}$, strep tagged APC/C was pre-incubated on ice with streptavidin-conjugated Alexa Fluor 700 (Thermo Fisher Scientific S21383) for 30 min. This was mixed with CDC20 and unlabelled Cyclin B1$^{NTD}$ at a 1:1:1 molar ratio. About 0.2 µM NCP147 was mixed with this APC/C solution at ratios of 1:0, 1:1, 1:2, 1:4, 1:8 and 0:1 in 6 µL volumes. The reactions were treated as with the CyclinB1 EMSAs and run on a 1% agarose gel made in 0.5X TBE and run in a 0.25X TBE running buffer. This was run for 90 min at 100 V.

For EMSAs of the NCP with APC3 loop, 0.5 µM of NCP147 was mixed with the APC3 loop construct at ratios of 1:0, 1:1, 1:2, 1:4, 1:6 and 1:8 in a 6 µL reaction volume. The reaction buffer used was 20 mM HEPES pH 7.5, 150 mM NaCl and 0.1 mM TCEP. Reactions were incubated on ice for 10 min before adding 5% (v/v) sucrose. A 5% polyacrylamide gel was pre-run for 60 min at 100 V in 0.25 XTBE at 4 °C before running the samples on the gel using the same conditions but for 90 min. The gels were imaged using a Typhoon FLA 9500 imager.

## Cross-linking mass spectrometry

About 1.4 µM of APC/C, CDC20 and Cyclin B1$^{NTD}$ were mixed to give a 1:1:1 molar ratio. 0.43 µM NCP was mixed with 3.4 µM of the APC/C-CDC20-Cyclin B1$^{NTD}$ mix and incubated on ice for 10 min. About 1.44 mM of DSSO (disuccinimidyl sulfoxide, A33545, (Thermo Scientific)) dissolved in DMSO was added to the proteins to give a final volume of 110 µL, this was incubated on ice for 1.5 h. The sample was quenched with 50 mM TRIS-HCl pH 8.0. After the crosslinking reaction, triethylammonium bicarbonate buffer (TEAB) was added to the sample at a final concentration of 100 mM. Proteins were reduced and alkylated with 5 mM tris-2-carboxyethyl phosphine (TCEP) and 10 mM iodoacetamide (IAA) simultaneously for 60 min in the dark and were digested overnight with trypsin at a final concentration of 50 ng/µL (Pierce). The sample was dried and peptides were fractionated with high-pH reversed-phase (RP) chromatography using the XBridge C18 column (1.0 × 100 mm, 3.5 µm, Waters) on an UltiMate 3000 HPLC system. Mobile phase A was 0.1% v/v ammonium hydroxide and mobile phase B was acetonitrile, 0.1% v/v ammonium hydroxide. The peptides were fractionated at 70 µL/min with the following gradient: 5 min at 5% B, up to 15% B in 3 min, for 32 min gradient to 40% B, gradient to 90% B in 5 min, isocratic for 5 min and re-equilibration to 5% B. Fractions were collected every 100 s, SpeedVac dried and pooled into 12 or 8 samples for MS analysis. LC-MS analysis was performed on an UltiMate 3000 or a Vanquish Neo UHPLC system coupled with the Orbitrap Ascend mass spectrometer (Thermo Scientific). Each peptide fraction was reconstituted in 30 µL 0.1% TFA and 15 µL were loaded to the Acclaim PepMap 100, 100 µm × 2 cm C18, 5 µm trapping column at 10 µL/min flow rate of 0.1% TFA loading buffer (U3000) or the PEPMAP 100 C18 5 µm 0.3 × 5 mm 1500 Bar (Neo). Peptides were then subjected to a gradient elution on a 25 cm capillary column (Waters, nanoE MZ PST BEH130 C18, 1.7 µm, 75 µm × 250 mm) connected to the EASY-Spray source at 45 °C with an EASY-Spray emitter (Thermo, ES991). Mobile phase A was 0.1% formic acid and mobile phase B was 80% acetonitrile, 0.1% formic acid. The separation method at a flow rate of 300 nL/min was an 80 min gradient from 5–35% B. Precursors between 380–1400 m/z and charge states 3–8 were selected at 120,000 resolution in the top speed mode in 3 s and were isolated for stepped HCD fragmentation (collision energies % = 21, 27, 34) with quadrupole isolation width 1.6 Th, Orbitrap detection with 30,000 resolution and 70 ms maximum injection time. Targeted MS precursors were dynamically excluded for further isolation and activation for 45 s with 10 ppm mass tolerance. Identification of crosslinked peptides was performed in Proteome Discoverer 3 (Thermo) with the MS Annika search engine node for DSSO/+158.004 Da (K). Precursor and fragment mass tolerances were 10 ppm and 0.02 Da, respectively, with a maximum of four trypsin missed cleavages allowed. Carbamidomethyl at C was selected as static modification and oxidation of M as dynamic modification. Spectra were searched against a FASTA file containing the sequences of the proteins in each complex concatenated with 1000 random UniProt *E. Coli* sequences as negative control. Crosslinked peptides were filtered at FDR <0.01 separately for intra/inter-links using a target-decoy database search.

The mass spectrometry proteomics data have been deposited to the ProteomeXchange Consortium via the PRIDE partner

repository with the dataset identifiers PXD046458 and PXD052677 (Perez-Riverol et al, 2022).

## Live-cell imaging and image analysis

Measurements of mitotic timing were obtained using Differential Interference Contrast (DIC) imaging on a Nikon Eclipse microscope (Nikon) equipped with a 20× 0.75 NA objective (Nikon), a Flash4.0 CMOS camera (Hamamatsu). Single plane images were taken every 3 min for 24 h using Micromanager software (μManager). Where indicated, cells were treated with paclitaxel.

Images and quantifications of Cyclin B1 and Securin, excluding FCS and FCCS (see below) were obtained on a Marianas confocal spinning-disk microscope system (Intelligent Imaging Innovations, Inc.) equipped with a laser stack for 445 nm/488 nm/514 nm/561 nm lasers, a 63 × 1.2 NA objective (Carl Zeiss), a Flash4 CMOS camera (Hamamatsu) and Slidebook 6 software (Intelligent Imaging Innovation, Inc.).

For Fig. 1, eight Z stacks (Step size = 1 μm) were taken every 30 s, for 90 min, using 20% 488 nm laser power and 10% 647 nm laser power, for 50 ms exposure, 1 × 1 binning. Image stacks were maximum projected over the z-axis and mean fluorescence intensity at different subcellular location was quantified. After background subtraction, each measurement was normalised on the value at 20 frames prior to anaphase and corrected for bleaching using the measurement obtained in the same subcellular location in a sample treated with 50 μg/ml cycloheximide (Merk) and 10 μM MG132. Where indicated, cells were treated with MG132 or APCin and Tame for 30 min prior to filming.

For Fig. 2, cells were exposed for 30 min to 50 nM Cenp-E inhibitor and treated with 5 uM Reversine immediately before imaging. Ten Z stacks (Step size = 1 μm) were taken every 30 s, using 7% 488 nm for 150 ms, and 5% 647 nm laser power, for 100 ms exposure, 4 × 4 binning. After background subtraction, a region of interest containing the chromatids of polar chromosomes was manually selected and the Cyclin B1 signal was quantified inside and outside the DNA area. Each measurement was normalised to the value obtained at 25 frames prior to anaphase.

For Cyclin B1 and Securin line-profile assays, cells were exposed to MG132 for 30 min prior to filming. Fifty-one Z stacks (Step size = 0.2 μm) were taken at a single time point, using 10% 488 nm, 15% 561 nm and 5% 647 nm laser power, for 100 ms exposure, 1 × 1 binning. Image stacks were summed over the z-axis and the fluorescence intensity over a 10-pixel thick line going from centrosome to centrosome was quantified. After background subtraction, each measurement was scaled between the maximum and minimum values, set as 1 and 0, respectively. In case of the Cyclin B1$^{4E7E}$ clones the line-profile was obtained on the same image used to quantify Cyclin B1 degradation (see below), 10 min before anaphase.

For Cyclin B1-mScarlet and Securin-mScarlet degradation assays, ten Z stacks (Step size = 1 μm) were taken every minute, using 10% 488 nm, 10% 561 nm and 10% 647 nm laser power, for 50 ms exposure, 2 × 2 binning. Image stacks were maximum projected over the z-axis and raw integrated density of the whole cell or of specific subcellular locations was measured. After background subtraction, each measurement was either scaled between the value at 25 frames prior to anaphase and the minimum value, arbitrarily set as 1 and 0, or normalised to the value at 25 frames prior to anaphase.

In the quantification of Cyclin B1 degradation in Cyclin B1$^{4E7E}$ clones, Cyclin B1 levels were normalised to frame −10 prior to anaphase and bleaching-corrected to a straight-line function fitted between frame −30 and −25 prior to anaphase. To obtain the maximum degradation speed and time, we fitted a straight-line function using five points around the minimum of the first derivative of Cyclin B1 degradation data smoothed with a four-values averaging window. Maximum degradation speed is the slope of that function. To find the initial degradation point, data were smoothed with a four-values averaging window and then fitted to a straight-line function using a sliding five-values window. The initial degradation point is the initial value of the first five-value window whose slope is below −0.05. The initial degradation speed is the slope of the window. All image processing and analysis was performed using Fiji (ImageJ).

## Fluorescence correlation and cross-correlation spectroscopy

A Leica TCS SP8 confocal microscope (DMI8; Leica) integrated with wavelength-adjustable pulsed white light laser and Leica HyD SMD (single molecule detection) detectors was used for all FCS and FCCS measurements. The samples were imaged and measured using a Leica HC PL APO CS2 63x/1.20 water immersion with a pinhole size of 1 airy unit. For proteins tagged with mEmerald, the wavelength was set at 488 nm with a detection range of 505–540 nm, while the wavelength was set at 569 nm and a detection range of 580–625 nm for measuring proteins tagged with mEmerald.

Calibration of the instrument prior to each FCS and FCCS experiment and FCCS control experiments (positive and negative) were performed as described in Barbiero et al, 2022. The auto- and cross-correlation functions were fitted using a Leica LAS X SMD FCS module. Selection of fitting models, determination of diffusion coefficients, calculation of cross-correlation quotient (q) and dissociation constant (KD) were also performed as described in Barbiero et al, 2022.

## Chromosome spreads

Cells were treated for 3 h with 100 ng ml$^{-1}$ colcemid (GIBCO), trypsinized and recovered in a Falcon tube. The cell suspension was centrifuged for 3 min at 250 × g and resuspended in 5 ml of 75 mM KCl, added dropwise. After a 15 min incubation at 37 °C, ten drops of Carnoys Fixative (3:1 methanol:acetic acid) were added. Following a 5 min centrifugation at 200 × g, the cell pellet was resuspended in 5 ml of Carnoys Fixative. After 90 min at −20 °C, a second fixation was performed using 5 ml of Carnoys Fixative at room temperature for 15 min. Cells were centrifuged at 200 × g for 5 min, the supernatant removed, and the pellet resuspended in 200 μl of Carnoys Fixative. Spreads were performed by dropping the cells from a height of 30 to 40 cm onto wet slides in a wet chamber. Spreads were mounted and stained using Vectashied with DAPI (ReactoLab). Samples were imaged on a GE wide-field DeltaVision Elite microscope equipped with a 63x/1.2 NA Oil Objective and SoftWoRx Imaging software. The number of chromosomes per cell was counted using ImageJ software.

## Protein alignment

For the protein alignment in Fig. 2A, we extracted the arginine anchors of the following proteins: LANA-1 (uniprot Q9QR71), Sir3

BAH (P06701), CenpC (Q03188), Set8 (Q9NQR1), 53BP1 (Q12888), DNM3B (Q9UBC3), HMGN2 (P05204), Set1 (P38827), BAF (Q12824) and CHD1 (P32657). For alignment in Fig. 2B we used full-length Cyclin B sequences from different species: *H. sapiens* (P14635), *M. musculus* (P24860), *X. laevis* (P13350), *D. rerio* (Q9IB44), *C. elegans* (Q10653) and *S. pombe* (P10815). For the sequence alignment of the APC3 loop, the following homologous sequences were taken from ProViz: *H. sapiens* (P30260), *B. taurus* (A7Z061), *A. thaliana* (A0A178VWS3), *D. melanogaster* (Q9VS37), *G. gallus* (Q5ZK91), *X. tropicalis* (Q0P4V8), *C. elegans* (Q9N593), *S. cerevisiae* (P38042), *S. pombe* (P10505). All alignments were performed using MUSCLE (EMBL-EBI) with default settings and colour-coded matching the Clustal colour scheme.

## Statistics and figure assembly

Details on statistics are summarised in Appendix Table S2. Data analysis was performed with Python 3.7.0. Statistical analysis and plotting were performed with Prism 8 (GraphPad). Mitotic timing graphs were generated using the 'Superplots' pipeline (Lord et al, 2020). Figures were assembled using Adobe Illustrator (Adobe).

## Data availability

The structural coordinates for the NCP-CycB$^{NTD}$ and NCP-APC3 loop$^{375-381}$ have been deposited in the Protein Data Bank (https://www.rcsb.org/) with the accession numbers: PDB-ID 9FH9 and 9FGQ, respectively. The specific hyperlinks are https://www.rcsb.org/structure/9FH9 and https://www.rcsb.org/structure/9FGQ for NCP-CycB$^{NTD}$ and NCP-APC3 loop$^{375-381}$. The EM data were deposited in the Electron Microscopy Data Bank (https://www.ebi.ac.uk/emdb/) with the accession codes: EMD-50443 and EMD-50416. The specific hyperlinks are https://www.ebi.ac.uk/emdb/EMD-50443 and https://www.ebi.ac.uk/emdb/EMD-50416 for NCP - Cyclin B1 and NCP-APC3 loop, respectively. The mass spectrometry proteomics data are deposited in the ProteomeXchange Consortium via the PRIDE partner repository (https://www.ebi.ac.uk/pride/) with the identifiers PXD046458 and PXD052677.

The source data of this paper are collected in the following database record: biostudies:S-SCDT-10_1038-S44318-024-00194-2.

## Peer review information

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

## Acknowledgements

We thank Fay Cooke and Anja Hagting, for early work on the spatial regulation of Cyclin B1 degradation, and Martina Barbiero for the initial FCS data on Cyclin B1. We thank Prof. Stephen Jackson for sharing a RPE-1 TP53$^{-/-}$ cell line, and Jing Yang and Ziguo Zhang from David Barford's laboratory for their help with the APC/C baculovirus generation. We thank Nora Cronin and the London Consortium for Cryo-EM (LonCEM, The Francis Crick Institute) for cryo-EM data collection on the NCP$^{APC3loop}$ structure. We thank all the members of the Pines and Alfieri groups for helpful discussions. We gratefully acknowledge the support of the ICR Core facilities, in particular, light microscopy and flow cytometry. LC, SV, CC and JP were supported by an Investigator Award from Wellcome (209470/Z/17/Z), RY, RM and CA were supported by a Sir Henry Dale Fellowship 215458/Z/19/Z. RY and RM were also supported by the Institute of Cancer Research (ICR - GFR005X, GFR146X). CP was funded by an internship from Bologna University, MM was supported by a BSCB summer student grant, AA was supported by the In2Research programme.

## Author contributions

**Luca Cirillo**: Conceptualisation; Formal analysis; Supervision; Funding acquisition; Validation; Investigation; Visualisation; Methodology; Writing—original draft; Writing—review and editing. **Rose Young**: Formal analysis; Validation; Investigation; Methodology; Writing—original draft; Writing—review and editing. **Sapthaswaran Veerapathiran**: Formal analysis; Investigation; Methodology. **Annalisa Roberti**: Formal analysis; Investigation; Methodology. **Molly Martin**: Resources; Investigation. **Azzah Abubacar**: Resources; Investigation. **Camilla Perosa**: Resources; Investigation. **Catherine Coates**: Resources. **Reyhan Muhammad**: Resources; Formal analysis; Investigation. **Theodoros I Roumeliotis**: Formal analysis; Investigation; Methodology. **Jyoti S Choudhary**: Formal analysis;

Investigation; Methodology. **Claudio Alfieri**: Formal analysis; Supervision; Funding acquisition; Validation; Writing—original draft; Project administration; Writing—review and editing. **Jonathon Pines**: Conceptualisation; Formal analysis; Supervision; Funding acquisition; Validation; Writing—original draft; Project administration; Writing—review and editing.

Source data underlying figure panels in this paper may have individual authorship assigned. Where available, figure panel/source data authorship is listed in the following database record: biostudies:S-SCDT-10_1038-S44318-024-00194-2.

## Disclosure and competing interests statement

The authors declare no competing interests.

# Expanded View Figures

**Figure EV1. Characterisation of RPE-1 CyclinB1-mEmerald⁺ᐟ⁺ and APC8-mScarlet⁺ᐟ⁺ cells.**

(A) Representative fluorescence confocal image of RPE-1 CyclinB1-mEmerald⁺ᐟ⁺; APC8⁺ᐟ⁺ cells in interphase. The scale bar corresponds to 20 μm. (B) Representative fluorescence confocal images over time of a CyclinB1-mEmerald⁺ᐟ⁺; APC8⁺ᐟ⁺ cell progressing through mitosis. Time is expressed as mm:ss. The scale bar corresponds to 10 μm. (C) Dot plots of the mitotic timing of parental RPE-1, RPE-1 APC8-mScarlet⁺ᐟ⁺, RPE-1 CyclinB1-mEmerald⁺ᐟ⁺ and APC8-mScarlet⁺ᐟ⁺ cells, untreated (left) or treated with 100 nM paclitaxel (right). Each small dot represents one cell and large dots represent the median of independent experiments: $n \geq 83$ cells per condition, $N = 3$ independent experiments. Numbers on the graphs indicate the percentage of cells completing mitosis during the time of observation. Clones marked in red are the ones selected for all following experiments. (D) Top, growth curve of RPE-1 (black), RPE-1 APC8-mScarlet⁺ᐟ⁺(orange), RPE-1 CyclinB1-mEmerald⁺ᐟ⁺ and APC8-mScarlet⁺ᐟ⁺(red), $N = 3$ experiment. Mean ± standard deviation are plotted. Bottom, dot plot of the chromosome number of parental RPE-1, RPE-1 APC8-mScarlet⁺ᐟ⁺, RPE-1 CyclinB1-mEmerald⁺ᐟ⁺ and APC8-mScarlet⁺ᐟ⁺ cells. Each dot represents one chromosome spread: $n \geq 34$ spreads per condition, $N = 2$. (E) Representative anti-APC8, anti-APC4 and β-Tubulin immunoblot of cell lysates from parental RPE-1, RPE-1 APC8-mScarlet⁺ᐟ⁺, RPE-1 CyclinB1-mEmerald⁺ᐟ⁺ and APC8-mScarlet⁺ᐟ⁺ cells before and after immunodepleting APC4, compared with control immunodepletion with IgG. (F) Bar graphs representing the quantification of the immunoblot in (panel E). $N = 2$ independent experiments. Mean ± standard deviation are plotted. (G) Graph representing the autocorrelation function of APC8-mScarlet over time in the nucleus. (H) Quantification of normalised Cyclin B1 fluorescence levels over time measured by spinning-disk fluorescence microscopy in RPE-1 CyclinB1-mEmerald⁺ᐟ⁺ cells compared to RPE-1 CyclinB1-mEmerald⁺ᐟ⁺; RPE-1 APC8-mScarlet⁺ᐟ⁺, RPE-1 CyclinB1-mEmerald⁺ᐟ⁺ cells. $n \geq 9$ cells per condition, $N = 3$ independent experiments. Mean ± standard deviation are plotted. Figure EV1 – Supplementary text to determine whether APC8-mScarlet is properly incorporated into the APC/C, we reasoned that immunoprecipitating an APC/C subunit would result in a depletion of APC8 levels in the lysate. Measuring the levels of APC8 immunodepletion following immunoprecipitation of APC4 revealed no significant cgq RPE-1, RPE APC8-mScarlet⁺ᐟ⁺, and RPE APC8-mScarlet⁺ᐟ⁺; Cyclin B1-mEmerald⁺ᐟ⁺ cells, indicating that mScarlet does not interfere with APC8 incorporation into the APC/C.

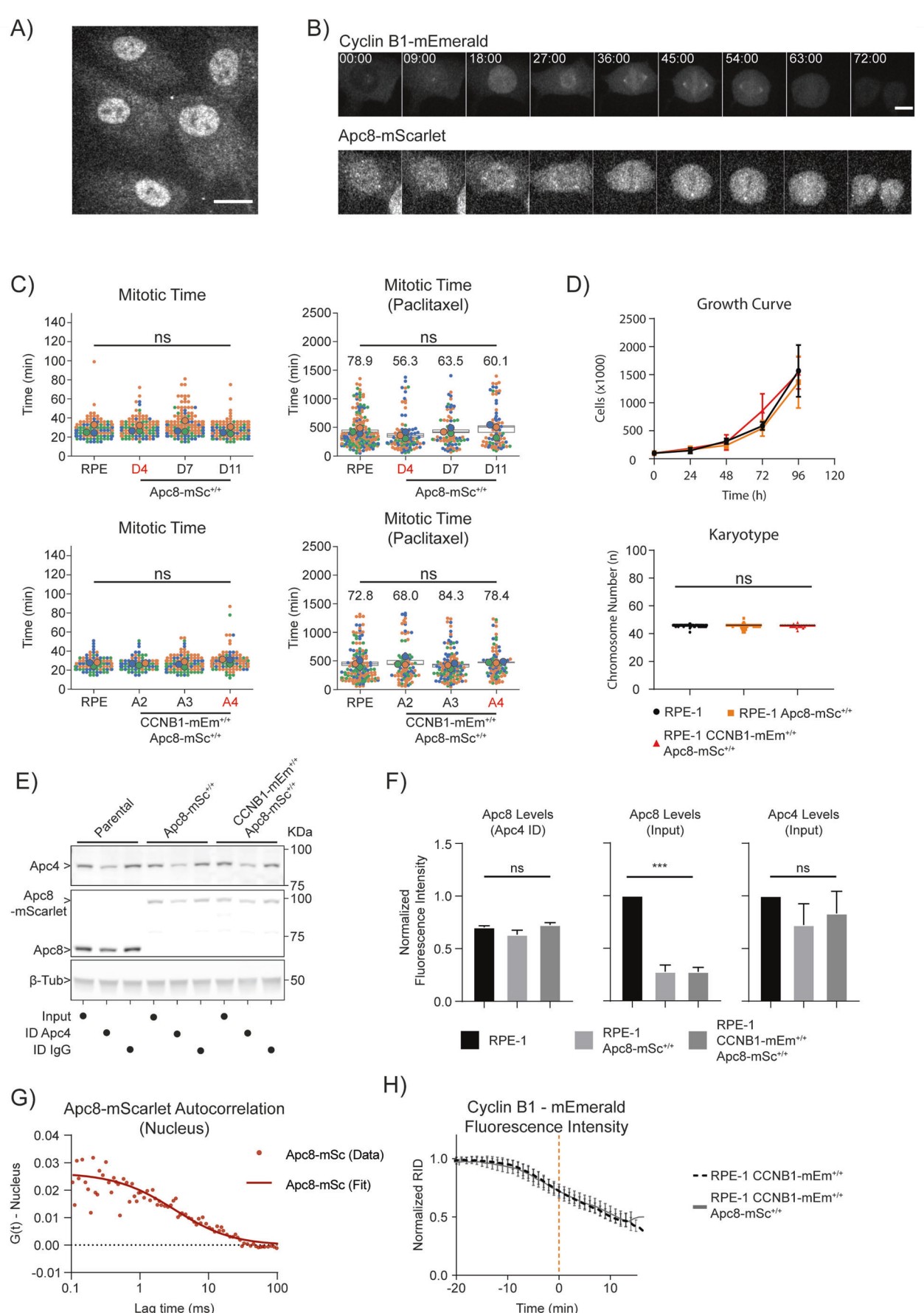

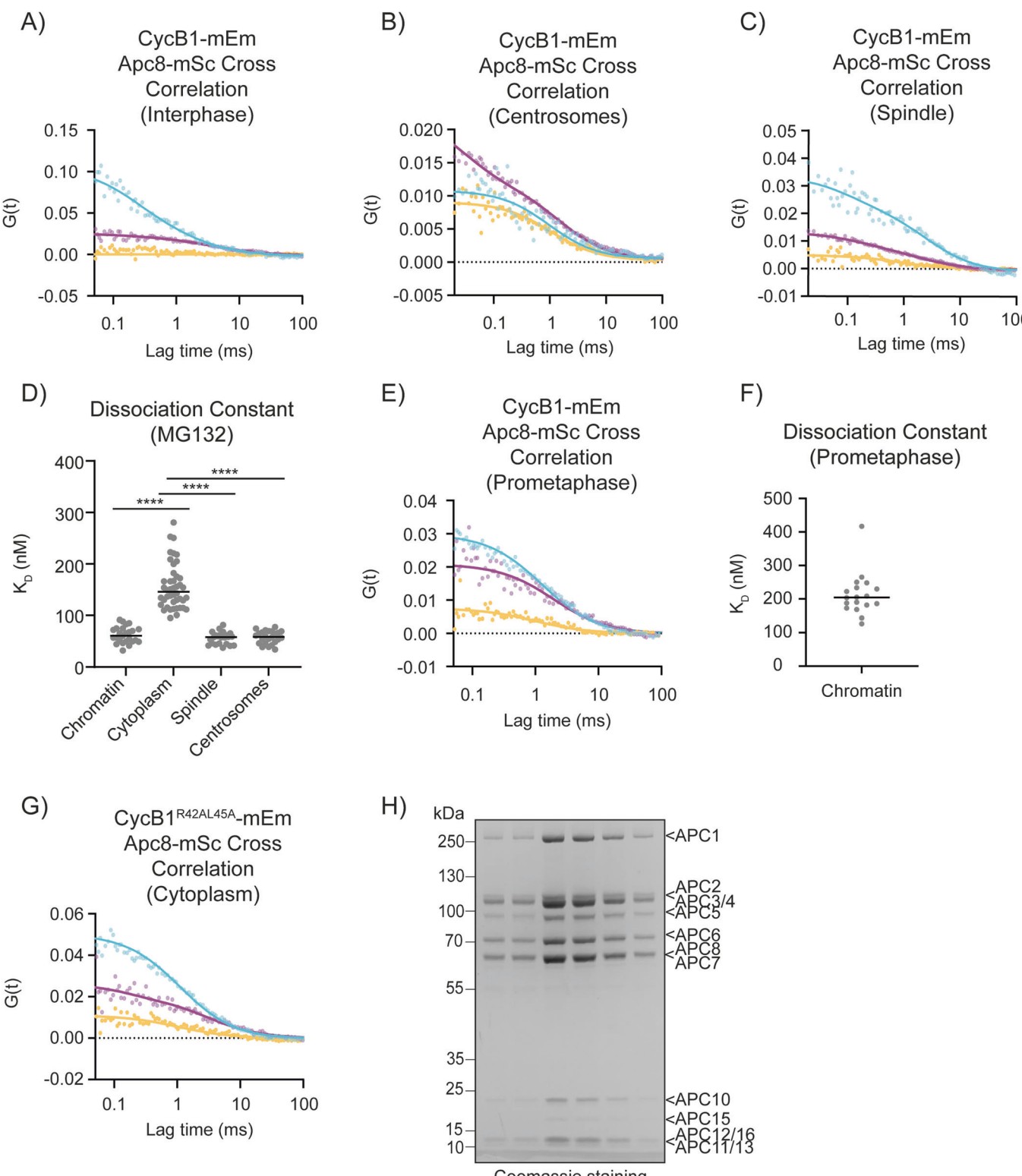

**Figure EV2.   FCCS of Cyclin B1-mEmerald and APC8-mScarlet.**

(**A–C, E, G**) Representative graphs of the autocorrelation function of mEmerald and mScarlet and the cross-correlation function between the two in RPE-1 CyclinB1-mEmerald$^{+/+}$; APC8-mScarlet$^{+/+}$ (**A–C, E**), and in RPE-1 APC8-mScarlet$^{+/+}$ cells ectopically expressing Cyclin B1$^{R45AL45A}$-mEmerald (**G**). (**D, F**) Dot plots representing the $K_D$ between endogenous Cyclin B1-mEmerald and APC8-mScarlet in metaphase cells following MG132 treatment (**D**) or in untreated prometaphase cells (**F**). $n \geq 18$ cells per condition, $N = 3$ independent experiments. (**H**) Fractions from a ResourceQ elution of the APC/C purification. Run on a 4–12% Bis-Tris SDS-PAGE gel.

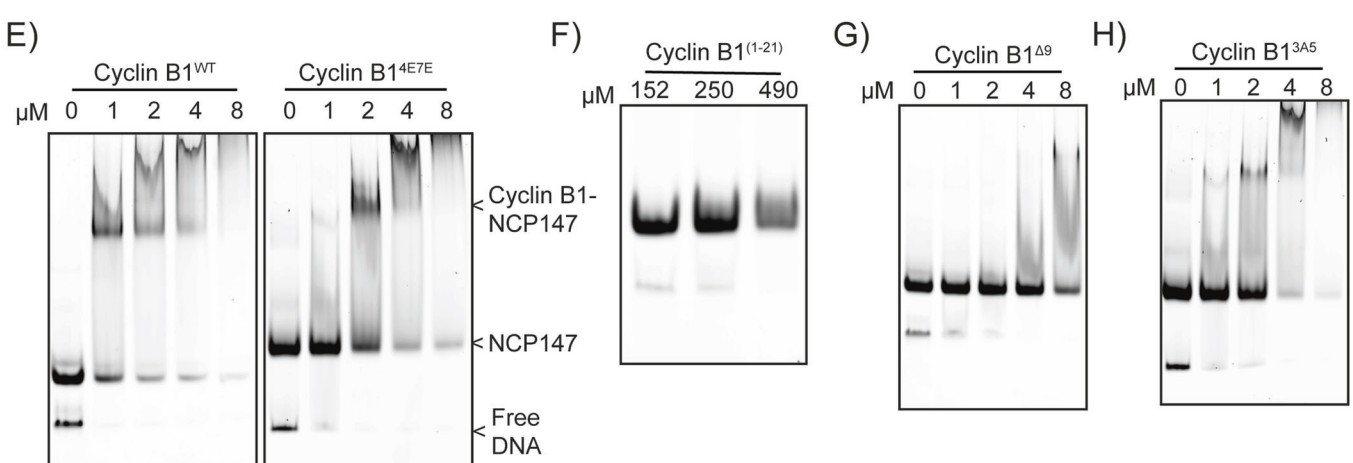

◀ **Figure EV3. Cyclin B1 binding to nucleosomes.**

(A) Representation of heteromeric crosslinks of the APC/C-CDC20-Cyclin B1^{NTD} complex with the NCP. (B) Representation of the self-crosslinks of individual APC/C subunits, CDC20, Cyclin B1^{NTD} and the four histones. (C) Representation of heteromeric crosslinks of the APC/C with the NCP. (D) Representation of self-crosslinks of individual APC/C subunits and histones. (E–H). EMSA of NCP147 with the indicated variant of Cyclin B1. $N = 3$ independent experiments.

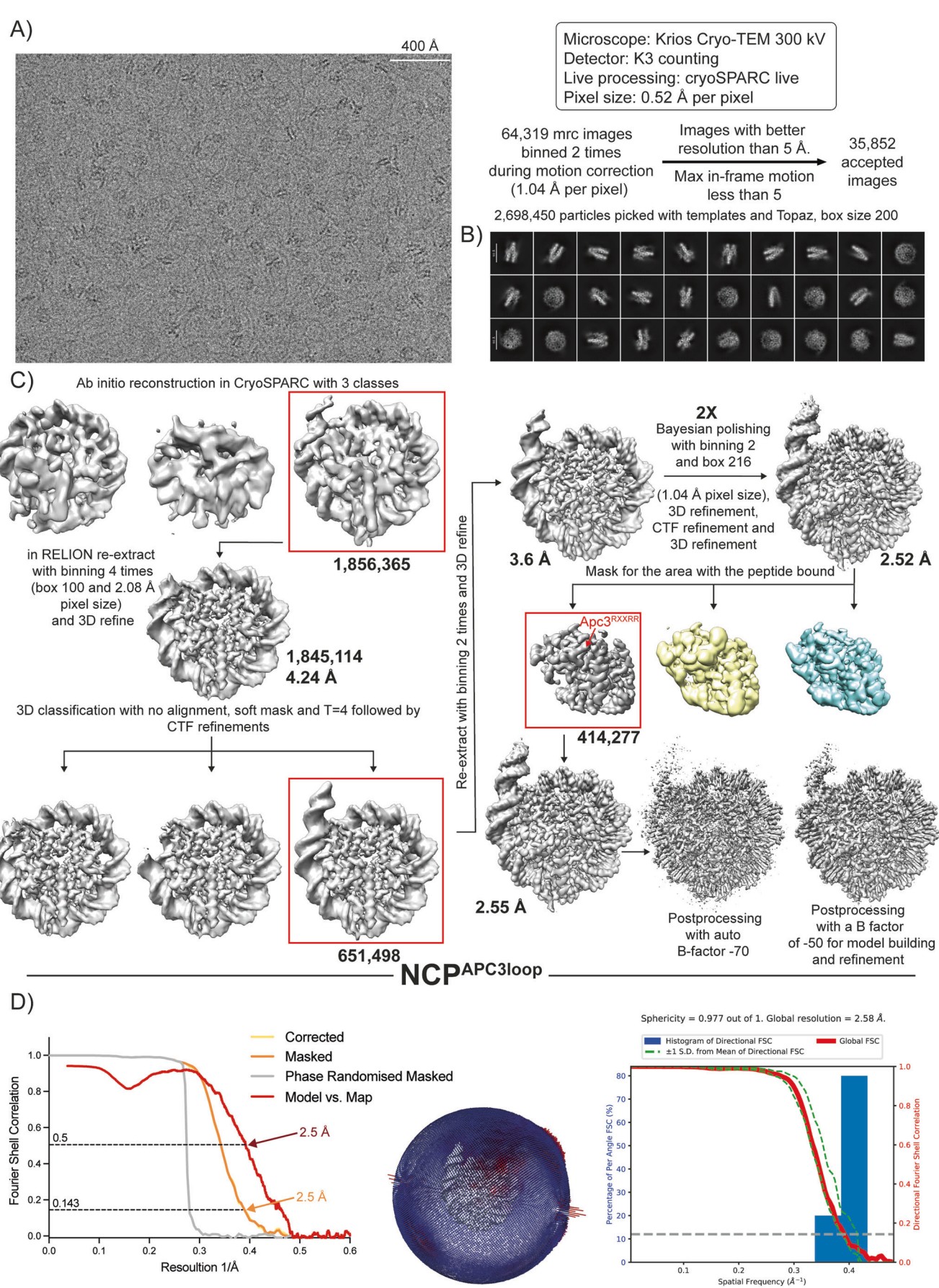

**Figure EV4. Cryo-EM analysis of the nucleosome core particle in complex with APC3 loop[177-446].**

(A–C) Workflow showing a representative micrograph, the cryo-EM data collection parameters (A) and the single-particle analysis pipeline for the nucleosome core particle with the APC3 loop[177-446] (B, C). N. of particles at each classification step is indicated. (D) Fourier shell correlation (FSC) curves, angular distribution plot and plot of the directional FSC that represent a measure of directional resolution anisotropy for all the reconstructions are shown. Directional FSC and sphericity determination was performed with the 3DFSC software (Tan et al, 2017).

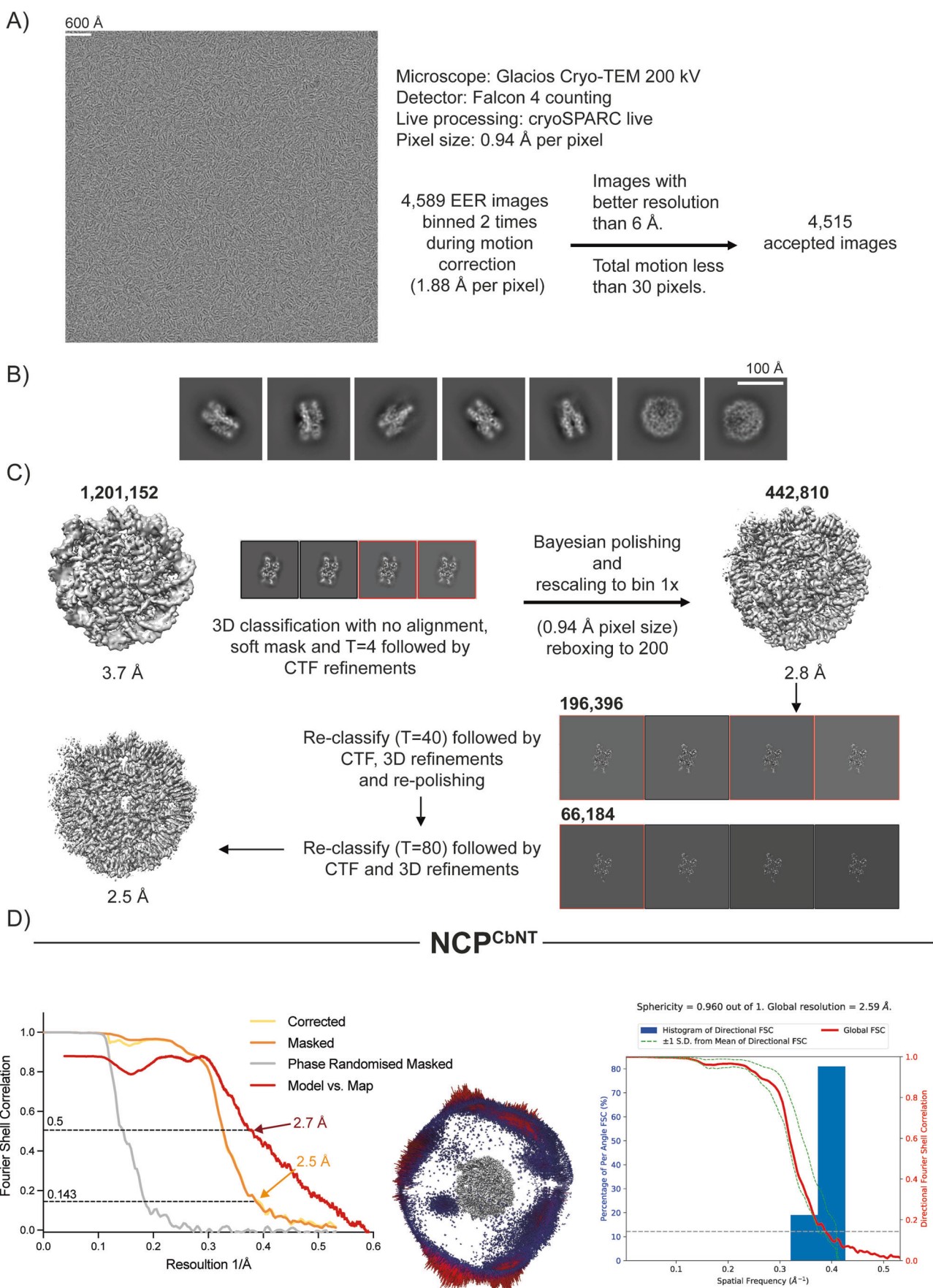

◀ **Figure EV5. Cryo-EM analysis of the nucleosome core particle in complex with Cyclin B1 N-terminus.**

(**A–C**) Workflow showing a representative micrograph, the cryo-EM data collection parameters (**A**) and the single-particle analysis pipeline for the nucleosome core particle in complex with Cyclin B N-terminus (NCP^CbNT) (**B, C**). N. of particles at each classification step is indicated. (**D**) Fourier shell correlation (FSC) curves, angular distribution plot and plot of the directional FSC that represent a measure of directional resolution anisotropy for all the reconstructions are shown. Directional FSC and sphericity determination was performed with the 3DFSC software.

