## [Peer Review File · The EMBO Journal]

Spatial control of the APC/C ensures the rapid degradation of Cyclin B1

Luca Cirillo, Rose Young, Sapthaswaran Veerapathiran, Annalisa Roberti, Molly Martin, Azzah Abubacar, Camilla Perosa, Catherine Coates, Reyhan Muhammad, Theo Roumeliotis, Jyoti Choudhary, Claudio Alfieri, and Jonathon Pines

Corresponding author(s): Jonathon Pines (Jon.Pines@icr.ac.uk) , Claudio Alfieri (claudio.alfieri@icr.ac.uk)

Review Timeline:

Submission Date:	12th Jan 24
Editorial Decision:	16th Feb 24
Revision Received:	30th May 24
Editorial Decision:	11th Jul 24
Revision Received:	22nd Jul 24
Accepted:	26th Jul 24

Editor: Hartmut Vodermaier

Transaction Report:

Dr. Jonathon Pines
The Institute of Cancer Research
London
United Kingdom

16th Feb 2024

Re: EMBOJ-2024-116650
Spatial control of the APC/C ensures the rapid degradation of Cyclin B1

Dear Jon,

Thank you for submitting your manuscript on spatial control of APC/C activity for our consideration. Three expert referees have now evaluated it and returned the below-copied reports. Since all of them appreciate the comprehensive and multidisciplinary approach, as well as the potential importance of your findings, we would be happy to consider a revised version further for EMBO Journal publication. The referees do nevertheless bring up a number of well-taken major concerns that would need to be answered prior to acceptance, many of them revolving around the nucleosome interactions of APC/C and cyclin B and their relative functional contribution, and also around the possibility of alternative interpretations.

Since it is our policy to aim for a single round of major revision, it will be important to diligently respond to each referee point at the time of resubmission, and I would therefore encourage you to contact me with a preliminary point-by-point response already during the early stages of your revision work, in order to clarify how key issues may be solved and for us to agree on a revision plan. We would also be open to extension of the default three-months revision period if needed; our 'scooping protection' (meaning that competing work appearing elsewhere in the meantime will not affect our considerations of your study) would of course remain valid also throughout such an extension.

Further information on preparing, formatting and uploading a revised manuscript can be found below and in our Guide to Authors. Thank you again for the opportunity to consider this work for The EMBO Journal, and I look forward to hearing from you in due time.

With kind regards,

Hartmut

4) Each main and each Expanded View (EV) figure should be uploaded as individual production-quality files (preferably in .eps,

.tif, .jpg formats). For suggestions on figure preparation/layout, please refer to our Figure Preparation Guidelines: <http://bit.ly/EMBOPressFigurePreparationGuideline>

9) Digital image enhancement is acceptable practice, as long as it accurately represents the original data and conforms to community standards. If a figure has been subjected to significant electronic manipulation, this must be clearly noted in the figure legend and/or the 'Materials and Methods' section. The editors reserve the right to request original versions of figures and the original images that were used to assemble the figure. Finally, we generally encourage uploading of numerical as well as gel/blot image source data; for details see: embopress.org/page/journal/14602075/authorguide#sourcedata

At EMBO Press, we ask authors to provide source data for the main manuscript figures. Our source data coordinator will contact you to discuss which figure panels we would need source data for and will also provide you with helpful tips on how to upload and organize the files.

In the interest of ensuring the conceptual advance provided by the work, we recommend submitting a revision within 3 months (16th May 2024). Please discuss the revision progress ahead of this time with the editor if you require more time to complete the revisions. Use the link below to submit your revision:

Link Not Available

Referee #1:

The timing of mitotic events is imperative for faithful chromosome segregation. Paramount to this process is the regulation of the Anaphase-Promoting Complex/Cyclome (APC/C) by the Spindle Assembly Checkpoint (SAC). While the APC/C targets many different substrates, the spatial regulation of its activity has remained relatively unclear. Several studies have shown that the APC/C has multiple substrates that bind to the nucleosome, and the APC/C itself is more active at chromatin. Specific to this manuscript, the authors focus on the recognition of Cyclin B by the APC/C. Interestingly, this group previously found that Cyclin B is removed from chromosomes first compared to the cytoplasm. In this manuscript, the authors propose an interesting model of how the APC/C and Cyclin B can both bind the nucleosome as a mechanism to facilitate the degradation of Cyclin B at the chromosomes. Overall, the manuscript is well written, and their model potentially explains several previous findings. Furthermore, their proposed mechanism could be broadly relevant to the dozens of APC/C substrates that may bind/localize to chromatin. However, a few points should be addressed prior to publication.

1. The primary concern of this reviewer is regarding the APC3-nucleosome interaction part of the proposed mechanism. The current data for this interaction seems premature compared to the rest of the manuscript. First, it is surprising the XL-MS data found only two peptides given the strength of the interaction considering multiple reports. Do the authors think that perhaps this is due to competition by having Cyclin B in the experiment? But if so, why are there not crosslinks to Cyclin B? Second, given the XL-MS does identify an interaction with H2A and H2B, the follow up on the finding is underwhelming as it stands. While the EMSA is convincing that there is an interaction, the specificity of the interaction in this experimental set up is unclear. The authors should include other proteins that are not expected to bind the NCP in these experiments. For example, an acid patch mutant NCP, APC3 loop mutants, or another apc subunit or loop could be used to show some selectivity. Such data would provide better support for the overall mechanism.

2. While the data is largely convincing for the Cyclin B-nucleosome interaction, there seems to be a slight discrepancy between the interaction data in the EMSA experiments compared to the degradation kinetics. As shown, the Cyclin B 4E7E does not bind that much weaker in the EMSA (Figure 4C and S3C) compared to WT, though it is called a significant decrease. However, its degradation (Figure 5B) seems slower than the Cyclin B delta 9 (Figure S6A), which binds much weaker (Figure 4C and S3E) compared to WT. While the rest of the data seem largely consistent, it seems like it is worth noting that there is a subtle disconnect between binding and degradation data.

3. The use of fluorescence cross correlation spectroscopy is particularly interesting to understand the spatial control of the Cyclin B degradation and determine the KD between it and the APC/C. Overall, its use was very interesting and informative. However, it is unclear to this reviewer why during the APCIN and TAME treatment the KD's "increased to a level similar to that found in the cytoplasm of either treated or untreated cells". Currently, the authors propose an alternative binding mode where Cyclin B-Cdk1-Cks binds APC/C through the Cks. This explanation would then suggest that Cyclin B in the cytoplasm is never bound as a substrate/through a D-box or the population is so small that it can't be distinguished through this method. Otherwise, shouldn't there be a decrease in the observed affinity? Perhaps, this issue is simply an experimental limitation, and the authors can help explain in a revised version.

Minor comments:

The color scheme of Figure 4J should be improved because the LANA-securin is hard to see.

Line 534 has a random "A" at the end of the paragraph.

Referee #2:

Comments on the manuscript entitled "Spatial control of the APC/C ensures the rapid degradation of Cyclin B1," submitted for publication in the EMBO Journal.

The manuscript adeptly examines the spatial dynamics of Cyclin B1 degradation during metaphase, essential for chromosome segregation and the prevention of aneuploidy. Employing state-of-the-art fluorescence cross-correlation spectroscopy (FCCS), in-depth in vitro biochemistry, and structural analysis, the study meticulously dissects the interaction between Cyclin B1 and the APC/C ubiquitin ligase, emphasising the chromosome as the primary interaction site, crucial for Cyclin B1's timely degradation.

Notably, the authors present a novel concept of direct nucleosome association by Cyclin B1 and APC/C subunits, offering new insights into their attachment to mitotic chromosomes. The paper further provides evidence for the physiological significance of this interaction, showing that mutations in Cyclin B1's nucleosome binding motif delay degradation and lead to aneuploidy, thus underlining the vital role of spatial regulation in genomic stability.

These insights contribute significantly to our understanding of mitotic regulation and the wider implications of spatially controlled protein degradation for cell cycle progression and genetic stability. The manuscript is also commendably written, with logical organisation and clarity that make it accessible and engaging to read.

Despite the strengths mentioned, there are several major concerns that prevent me from supporting its publication in its current state. In the following sections, I will discuss my concerns and suggest areas for improvement for this paper.

Major Concerns:

1. The utilisation of FCCS to study the interaction of Cyclin B with APC/C (Figure 1) offers direct evidence of spatial regulation within cells. However, the sole reliance on endogenously tagged APC8, which exhibits notably lower expression and uncertain chromosomal localisation, may not accurately represent the localisation and interaction dynamics of the entire APC/C. Given the reported diversity in localisation patterns and the potential functional independence of APC/C subunits, additional subunits like Apc3, which is claimed to bind directly to nucleosomes, should be examined. Furthermore, the characterisation of the APC8-tagged cell line seems incomplete. Thorough genomic validation and confirmation of integration into the APC/C complex via co-immunoprecipitation would reinforce these claims. The logic behind APC4 depletion in Supplementary Figure S1E and F needs clarification.

2. The hypothesis that APC/C and Cyclin B1 interact directly with nucleosomes (Figures 3-5) is intriguing but lacks sufficient support. For APC/C, a mutation analysis is warranted. Regarding Cyclin B, since the importance of basic patches in the N-terminus of Cyclin B for its chromosomal localisation was already shown in the cited paper, it is critical to determine whether it is indeed mediated by the direct nucleosome interactions, not through other DNA binding or other DNA binding proteins.

3. Using endogenous Cyclin B1-mEmerald to assess the degradation kinetics of ectopically expressed Cyclin B mutants (Figure 5) is a smart approach. However, the delayed degradation observed in mutants could indicate minor recognition or ubiquitination defects by APC/C. This point is crucial for the authors' claim regarding the necessity of chromatin-binding for efficient Cyclin B degradation and should be tested in *in vitro* binding or ubiquitination assays.

A comparative analysis of Cyclin B1 degradation in different subcellular compartments, as done for the 4E7E mutant (Figure 7D), would be insightful, particularly in light of the significant expression differences between the mutant and wild-type Cyclin B1.

4. The physiological relevance of spatial regulation on Cyclin B1 degradation is of significant interest. The assertion that the 4E7E mutant increases aneuploidy requires further evidence. Considering the notably lower expression of the mutant protein, an induction study to confirm its specific contribution to mitotic chromosome segregation is necessary. Otherwise, the authors should temper their claims.

5. The manuscript frequently omits statistical analysis (i.e., p-values) for observed differences in fluorescent intensity plots (Figures 1B-D, 2B, 4I, J, 5A, B, etc.). Although not every data point requires analysis, key time points where differences are highlighted should be statistically validated.

Minor Concerns and Corrections:

1. Line 60 lacks a reference for Nek2A; Hames et al., 2001 should be cited.
2. Line 80's use of "missing" is incorrect, given the substantial direct evidence for spatial regulation of the APC/C.
3. Figures 1B and 2B need graphical adjustments for clarity due to overlapping lines and error bars.
4. The figure legend requires an explanation for SiR-DNA.
5. Line 157 calls for additional relevant data and immunoblots to be provided.
6. Line 166 questions the low APC8 levels shown in the blot.
7. The rationale in lines 170-173 is not clear.
8. Figure S1E lacks a loading control.
9. Figure S1F is incorrectly labelled as "Cdc23-mSc."
10. Line 180 would benefit from including "chromosomes" in the section title.
11. Figure 2A should include magnified images of polar chromosomes at additional time points.
12. Line 182 should be rephrased for clarity to reflect the absence of cross-correlation in G2 phase cells and the strong interaction in metaphase.
13. Figure 5 requires representative live-cell imaging images.
14. Line 430 poses a question: Is the homozygous mutation lethal?
15. Figure 7G's axes should be switched for consistency with Figures 7E and F.
16. Line 546 should also cite Fujimitsu et al., 2016

Referee #3:

As the master regulator of mitotic progression, the activity of cyclin B1-CDK1 has to be tightly regulated. Degradation of cyclin B1 is required for mitotic exit and is under the control of the spindle checkpoint. After all chromosomes are properly attached to the spindle, cyclin B1 is rapidly ubiquitinated by the anaphase-promoting complex/cyclosome (APC/C) and degraded by the proteasome. Earlier studies from the Pines group and others have shown that cyclin B1 degradation starts at mitotic chromosomes at the metaphase-anaphase transition. The current study now provides insight into the molecular mechanism of such spatiotemporal regulation of cyclin B1 degradation.

Combining live-cell imaging, biochemical assays, and cryo-electron microscopy, Cirillo et al. show that cyclin B1 binds to mitotic chromosomes through an interaction between its N-terminal arginine anchor motif and the acidic patch on nucleosomes. Likewise, a loop region of APC/C also interacts with the acidic patch of nucleosomes. Mutation of the nucleosome-binding motif of cyclin B1 reduces its interaction with chromosomes, delays its degradation, and causes chromosome missegregation. Overall, the experiments are well designed with the necessary controls. Both loss- and gain-of function mutants are tested, with expected results. The technical quality of most results is high. The multidisciplinary approach is commendable. Publication of this excellent study is recommended provided that the authors address the following points.

Major points

(1) The authors did not provide an explanation as to why the binding of both cyclin B1 and APC/C to the acidic patch of nucleosomes stimulate their interactions and cyclin B1 ubiquitination. Do they compete for the same binding site on a given mono-nucleosome? Do oligo-nucleosomes bridge an interaction between cyclin B1 and APC/C? Do binding of both cyclin B1 and APC/C to oligo-nucleosomes simply increase their local concentrations? If the stimulation is due to local concentration effects, why cannot the spindle have the same stimulatory effect as both cyclin B1 and APC/C are enriched on the spindle? To begin to address these questions, the authors should test whether nucleosomes can stimulate cyclin B1 ubiquitination by APC/C using reconstituted APC/C ubiquitination assays *in vitro*.

(2) An alternative explanation of their results is that the checkpoint silencing activity is stronger on chromosomes. The APC/C pool bound to chromosomes is selectively activated first and preferably ubiquitinates chromosome-bound cyclin B1. This alternative model needs to be discussed, especially if the in vitro ubiquitination assay mentioned above cannot be done or produces negative results.

(3) The authors show that RPE-1 CCNB14E7E-mEmerald^{+/+} cells became aneuploid. How many clones were examined? Can this phenotype be observed with multiple clones? Does the spindle checkpoint work properly in these cells?

Dear Hartmut,

We wish to submit a revised version of our paper “Spatial control of the APC/C ensures the rapid degradation of Cyclin B1” for publication in The EMBO Journal. The referees were most concerned that we strengthen our data on the interaction between APC3 and nucleosomes, while referees 2 and 3 questioned the effect of the lower levels of Cyclin B1 in our mutant cell lines. We have performed extensive experiments to address these concerns. In support of a direct interaction between the APC3 loop and the nucleosome acidic patch, we now show in new Figure 3D and E that the APC3 loop alone is sufficient to bind to nucleosomes in an EMSA, in a manner dependent on the nucleosome acidic patch. Moreover, we determined the cryo-EM structure of the APC3 loop bound to the nucleosome, which revealed in new Figure 3F and G how arginine 380 directly engages the nucleosome acidic patch.

To distinguish between the effects of defective nucleosome binding and simply reducing Cyclin B1 levels we generated stable clones driving Cyclin B1 expression from an FRT-site in our Cyclin B1 arginine anchor mutant cell lines. We expressed either wild type Cyclin B1 or the arginine anchor mutant (4E7E) to a level comparable to normal endogenous Cyclin B1 for two weeks and found in new Figure Z that Cyclin B1^{4E7E} increased the percentage of aneuploid and polyploid cells compared to cells expressing wild type Cyclin B1. This result demonstrates that chromosome binding, rather than expression level, is responsible for the karyotypic aberrations observed in the arginine anchor mutant clones.

We have addressed all other referees' comments either experimentally or by rewriting the text as set out in the following point-by-point rebuttal.

Referee #1:

The timing of mitotic events is imperative for faithful chromosome segregation. Paramount to this process is the regulation of the Anaphase-Promoting Complex/Cyclin (APC/C) by the Spindle Assembly Checkpoint (SAC). While the APC/C targets many different substrates, the spatial regulation of its activity has remained relatively unclear. Several studies have shown that the APC/C has multiple substrates that bind to the nucleosome, and the APC/C itself is more active at chromatin. Specific to this manuscript, the authors focus on the recognition of Cyclin B by the APC/C. Interestingly, this group previously found that Cyclin B is removed from chromosomes first compared to the cytoplasm. In this manuscript, the authors propose an interesting model of how the APC/C and Cyclin B can both bind the nucleosome as a mechanism to facilitate the degradation of Cyclin B at the chromosomes. Overall, the manuscript is well written, and their model potentially explains several previous findings. Furthermore, their proposed mechanism could be broadly relevant to the dozens of APC/C substrates that may bind/localize to chromatin. However, a few points should be addressed prior to publication.

We thank the referee for their positive review of our study.

1. The primary concern of this reviewer is regarding the APC3-nucleosome interaction part of the proposed mechanism. The current data for this interaction seems premature compared to the rest of the manuscript. First, it is surprising the XL-MS data found only two peptides given the strength of the interaction considering multiple reports. Do the authors think that perhaps this is due to competition by having Cyclin B in the experiment? But if so, why are

there not crosslinks to Cyclin B? Second, given the XL-MS does identify an interaction with H2A and H2B, the follow up on the finding is underwhelming as it stands. While the EMSA is convincing that there is an interaction, the specificity of the interaction in this experimental set up is unclear. The authors should include other proteins that are not expected to bind the NCP in these experiments. For example, an acid patch mutant NCP, APC3 loop mutants, or another apc subunit or loop could be used to show some selectivity. Such data would provide better support for the overall mechanism.

We thank the referee for their suggestions and have significantly strengthened the data supporting the APC/C-nucleosome interaction. Firstly, we have repeated the cross-linking analysis in the absence of Cyclin B1 and obtained the same interaction. Secondly, a putative arginine anchor in the APC3 loop that is adjacent to the lysine that cross-links to the nucleosome is predicted by AlphaFold to bind into the acidic patch on nucleosomes. We mutated this arginine anchor and found that this prevents the APC3-dependent shifts on EMSA assays. Moreover, we used CryoEM to resolve the structure of a peptide containing the arginine anchor bound to nucleosomes. These data strongly support our conclusion that the APC3 loop binds to the nucleosome acidic patch and are presented in new Fig. 3 and new Figure S4 and Table S3.

2. While the data is largely convincing for the Cyclin B-nucleosome interaction, there seems to be a slight discrepancy between the interaction data in the EMSA experiments compared to the degradation kinetics. As shown, the Cyclin B 4E7E does not bind that much weaker in the EMSA (Figure 4C and S3C) compared to WT, though it is called a significant decrease. However, its degradation (Figure 5B) seems slower than the Cyclin B delta 9 (Figure S6A), which binds much weaker (Figure 4C and S3E) compared to WT. While the rest of the data seem largely consistent, it seems like it is worth noting that there is a subtle disconnect between binding and degradation data.

Although we agree with the reviewer that there is a difference in the behaviour of the mutants in the EMSA assay versus their degradation, it is perhaps to be expected given that these assays are very different (a measure of in vitro complex assembly versus in vivo ubiquitylation and recognition by the proteasome).

3. The use of fluorescence cross correlation spectroscopy is particularly interesting to understand the spatial control of the Cyclin B degradation and determine the KD between it and the APC/C. Overall, its use was very interesting and informative. However, it is unclear to this reviewer why during the APCIN and TAME treatment the KD's "increased to a level similar to that found in the cytoplasm of either treated or untreated cells". Currently, the authors propose an alternative binding mode where Cyclin B-Cdk1-Cks binds APC/C through the Cks. This explanation would then suggest that Cyclin B in the cytoplasm is never bound as a substrate/through a D-box or the population is so small that it can't be distinguished through this method. Otherwise, shouldn't there be a decrease in the observed affinity? Perhaps, this issue is simply an experimental limitation, and the authors can help explain in a revised version.

. It is likely that the cytoplasmic affinity we measure is experimentally limited and we cannot formally exclude that Cyclin B1 binds APC/C in a D-box dependent manner in the cytoplasm. We now mention this in the text (line 206). Nevertheless, because cytoplasmic binding affinity between cyclin B1 and APC/C is unchanged following APC/C inhibition or in cells expressing a D-box mutant of cyclin B1 or in cells in prometaphase (where cyclin B1

recognition by the APC/C is blocked by the MCC), our data strongly indicate that cytoplasmic D-box dependent binding is negligible.

Minor comments:

The color scheme of Figure 4J should be improved because the LANA-securin is hard to see. We have modified the colour scheme in Figure 4J

Line 534 has a random "A" at the end of the paragraph.

We have corrected these.

Referee #2:

Comments on the manuscript entitled "Spatial control of the APC/C ensures the rapid degradation of Cyclin B1," submitted for publication in the EMBO Journal.

The manuscript adeptly examines the spatial dynamics of Cyclin B1 degradation during metaphase, essential for chromosome segregation and the prevention of aneuploidy. Employing state-of-the-art fluorescence cross-correlation spectroscopy (FCCS), in-depth in vitro biochemistry, and structural analysis, the study meticulously dissects the interaction between Cyclin B1 and the APC/C ubiquitin ligase, emphasising the chromosome as the primary interaction site, crucial for Cyclin B1's timely degradation.

Notably, the authors present a novel concept of direct nucleosome association by Cyclin B1 and APC/C subunits, offering new insights into their attachment to mitotic chromosomes. The paper further provides evidence for the physiological significance of this interaction, showing that mutations in Cyclin B1's nucleosome binding motif delay degradation and lead to aneuploidy, thus underlining the vital role of spatial regulation in genomic stability.

These insights contribute significantly to our understanding of mitotic regulation and the wider implications of spatially controlled protein degradation for cell cycle progression and genetic stability. The manuscript is also commendably written, with logical organisation and clarity that make it accessible and engaging to read.

Despite the strengths mentioned, there are several major concerns that prevent me from supporting its publication in its current state. In the following sections, I will discuss my concerns and suggest areas for improvement for this paper.

We thank the referee for their positive review of our study.

Major Concerns:

1. The utilisation of FCCS to study the interaction of Cyclin B with APC/C (Figure 1) offers direct evidence of spatial regulation within cells. However, the sole reliance on endogenously tagged APC8, which exhibits notably lower expression and uncertain chromosomal localisation, may not accurately represent the localisation and interaction dynamics of the entire APC/C. Given the reported diversity in localisation patterns and the potential functional independence of APC/C subunits, additional subunits like Apc3, which is claimed to bind directly to nucleosomes, should be examined.

Furthermore, the characterisation of the APC8-tagged cell line seems incomplete. Thorough

genomic validation and confirmation of integration into the APC/C complex via co-immunoprecipitation would reinforce these claims. The logic behind APC4 depletion in Supplementary Figure S1E and F needs clarification.

We thank the reviewer for these comments. In addition to data showing that all the tagged APC8 is incorporated into the APC/C (Fig. S1 E, G), and that cells have control levels of APC/C activity (Fig. S1H) and SAC strength (Fig. S1 C), we have also tagged APC3, APC1 and APC4 and these all show the same diffuse overall distribution in cells as APC8. Thus, it is most likely that sub-populations of the APC/C are more important, as revealed by FCCS. We have included the sequencing results of clones A4 and D4 in the source data, and have added text to clarify the rationale behind Fig S1E and F.

2. The hypothesis that APC/C and Cyclin B1 interact directly with nucleosomes (Figures 3-5) is intriguing but lacks sufficient support. For APC/C, a mutation analysis is warranted. Regarding Cyclin B, since the importance of basic patches in the N-terminus of Cyclin B for its chromosomal localisation was already shown in the cited paper, it is critical to determine whether it is indeed mediated by the direct nucleosome interactions, not through other DNA binding or other DNA binding proteins.

We thank the referee for these comments and have significantly strengthened that evidence that the APC/C directly binds to nucleosomes: we have identified an arginine anchor in a loop of APC3; we show that mutating the arginine anchor prevents nucleosome binding; we have determined the cryo-EM structure of the APC3 arginine anchor bound to the nucleosome (Fig 3).

We are slightly unclear what evidence this referee requires to show that Cyclin B1 binds chromosomes directly through the acidic patch. We show that a Cyclin B1 arginine anchor mutant has markedly reduced binding to nucleosomes in an EMSA assay (Fig 4C), and that this mutation prevents chromosome binding in vivo, which can be rescued by a LANA peptide. Thus, our data show that the arginine anchor of Cyclin B1 is necessary and sufficient for nucleosome binding, although we do not exclude that other parts of cyclin B1 may also interact with the chromatin as described in Pfaff and King, 2013.

3. Using endogenous Cyclin B1-mEmerald to assess the degradation kinetics of ectopically expressed Cyclin B mutants (Figure 5) is a smart approach. However, the delayed degradation observed in mutants could indicate minor recognition or ubiquitination defects by APC/C. This point is crucial for the authors' claim regarding the necessity of chromatin-binding for efficient Cyclin B degradation and should be tested in in vitro binding or ubiquitination assays.

We thank the referee for this comment. We suggest that our data in Fig 6A largely excludes the possibility that the 4E7E mutation perturbs APC/C recognition because size exclusion chromatography analysis shows that cyclin B1 4E7E binding to APCC is indistinguishable from that of wild-type cyclin B1. Moreover, a Cyclin B1 mutant in which the first 9 amino acids were replaced with the LANA peptide (Fig 6C) display better degradation kinetics than Cyclin B1^{Δ9}, indicating that it is the lack of nucleosome interaction rather than defective APC/C recruitment that is responsible for the delayed degradation of Cyclin B1.

A comparative analysis of Cyclin B1 degradation in different subcellular compartments, as done for the 4E7E mutant (Figure 7D), would be insightful, particularly in light of the significant expression differences between the mutant and wild-type Cyclin B1.

Ideally, we would wish to do this, but we were unable to identify mutants of Cyclin B1 with altered localisation that also retained binding to Cdk1. We do not exclude that other subcellular compartments could contribute to cyclin B1 degradation, but our data argue that chromatin is the major site of Cyclin B1 ubiquitylation: when chromatin binding is perturbed this delays Cyclin B1 degradation throughout the cell.

4. The physiological relevance of spatial regulation on Cyclin B1 degradation is of significant interest. The assertion that the 4E7E mutant increases aneuploidy requires further evidence. Considering the notably lower expression of the mutant protein, an induction study to confirm its specific contribution to mitotic chromosome segregation is necessary. Otherwise, the authors should temper their claims.

We thank the referee for this comment and include new data to address this concern. We generated stable tetracyclin-inducible cell lines to increase the levels of Cyclin B1 in our 4E7E mutant cell lines and compared the effect of wild type Cyclin B1 to the arginine anchor mutant. Ectopic expression of wild type cyclin B1 had no significant effect on ploidy whereas increasing the levels of the 4E7E mutant caused a large increase in mitotic errors – both aneuploidy and polyploidy (new Fig. 7I). The likelihood is that delayed degradation of Cyclin B1 increases the amount of Cyclin B1 in late metaphase and anaphase, and this causes problems in chromosome segregation and cytokinesis.

5. The manuscript frequently omits statistical analysis (i.e., p-values) for observed differences in fluorescent intensity plots (Figures 1B-D, 2B, 4I, J, 5A, B, etc.). Although not every data point requires analysis, key time points where differences are highlighted should be statistically validated.

We have added these to the other statistical measures in the source data.

Minor Concerns and Corrections:

1. Line 60 lacks a reference for Nek2A; Hames et al., 2001 should be cited.

We have included this reference.

2. Line 80's use of "missing" is incorrect, given the substantial direct evidence for spatial regulation of the APC/C.

We have amended the sentence.

3. Figures 1B and 2B need graphical adjustments for clarity due to overlapping lines and error bars.

We now provide the source data for further clarity.

4. The figure legend requires an explanation for SiR-DNA.

This is now included in the Materials and Methods.

5. Line 157 calls for additional relevant data and immunoblots to be provided.

Please see response to this reviewer's Point 1.

6. Line 166 questions the low APC8 levels shown in the blot.

We have quantified the blot in fig S1F and describe this in the materials and methods. In addition, we now provide the raw data for both the western blot and its quantification.

7. The rationale in lines 170-173 is not clear.

We apologise for this and have added additional text to Fig S1.

8. Figure S1E lacks a loading control.

We now include a tubulin loading control.

9. Figure S1F is incorrectly labelled as "Cdc23-mSc."

We have corrected this.

10. Line 180 would benefit from including "chromosomes" in the section title.

Thank you, we have amended this.

11. Figure 2A should include magnified images of polar chromosomes at additional time points.

We now include a magnified chromosome for all time points.

12. Line 182 should be rephrased for clarity to reflect the absence of cross-correlation in G2 phase cells and the strong interaction in metaphase.

We have clarified the text.

13. Figure 5 requires representative live-cell imaging images.

We have provided images of these cells in Figure 4H.

14. Line 430 poses a question: Is the homozygous mutation lethal?

We successfully generated three homozygous Cyclin B14E7E cell lines when we knocked-out p53; therefore, it is likely that cells carrying a homozygous Cyclin B14E7E mutation undergo a p53-dependent cell cycle arrest.

15. Figure 7G's axes should be switched for consistency with Figures 7E and F.

Figure 7G axis are consistent with the degradation curves showed in panels C and D.

16. Line 546 should also cite Fujimitsu et al., 2016

We have included this reference.

Referee #3:

As the master regulator of mitotic progression, the activity of cyclin B1-CDK1 has to be tightly regulated. Degradation of cyclin B1 is required for mitotic exit and is under the control of the spindle checkpoint. After all chromosomes are properly attached to the spindle, cyclin B1 is rapidly ubiquitinated by the anaphase-promoting complex/cyclosome (APC/C) and degraded by the proteasome. Earlier studies from the Pines group and others have shown that cyclin B1 degradation starts at mitotic chromosomes at the metaphase-anaphase transition. The current study now provides insight into the molecular mechanism of such spatiotemporal regulation of cyclin B1 degradation.

Combining live-cell imaging, biochemical assays, and cryo-electron microscopy, Cirillo et al. show that cyclin B1 binds to mitotic chromosomes through an interaction between its N-terminal arginine anchor motif and the acidic patch on nucleosomes. Likewise, a loop region of APC/C also interacts with the acidic patch of nucleosomes. Mutation of the nucleosome-binding motif of cyclin B1 reduces its interaction with chromosomes, delays its degradation, and causes chromosome missegregation. Overall, the experiments are well designed with the necessary controls. Both loss- and gain-of-function mutants are tested, with expected results. The technical quality of most results is high. The multidisciplinary approach is commendable. Publication of this excellent study is recommended provided that the authors address the following points.

Major points

(1) The authors did not provide an explanation as to why the binding of both cyclin B1 and APC/C to the acidic patch of nucleosomes stimulate their interactions and cyclin B1 ubiquitination. Do they compete for the same binding site on a given mono-nucleosome? Do oligo-nucleosomes bridge an interaction between cyclin B1 and APC/C? Do binding of both cyclin B1 and APC/C to oligo-nucleosomes simply increase their local concentrations? If the

stimulation is due to local concentration effects, why cannot the spindle have the same stimulatory effect as both cyclin B1 and APC/C are enriched on the spindle? To begin to address these questions, the authors should test whether nucleosomes can stimulate cyclin B1 ubiquitination by APC/C using reconstituted APC/C ubiquitination assays in vitro.

We have extensively tested the effect of nucleosomes in in vitro ubiquitylation assays. We have tested many conditions, including varying ionic strength, concentrations of E2s, mono and di-nucleosomes, order of additions experiments, phosphatase-treated and untreated APC/C, wild type and N-terminal mutated CDC20 and CDH1, amongst others, but the sheer complexity of the assay means that we cannot control for all possible parameters. Previous studies have observed that chromatin-bound APC/C displays increased ubiquitylation activity when purified from mitotic HeLa cells (Sivakumar et al., 2014); therefore, we believe the situation in vivo is more complex than what we have thus far recapitulated with insect-purified APC/C. It is always possible that we are missing crucial interactions between kinases and phosphatases, or indeed other PTMs such as SUMO.

In our discussion we present three models to explain the increased ubiquitylation activity at chromosomes: 1) Polyanion dependent stimulation of APC/C activity (Mizrak and Morgan, 2019) (line 584); 2) Increased local concentration of both Cyclin B1 and APC/C (line 564); 3) Allosteric increase in APC/C activity through interaction with the APC3 loop (line 564).

(2) An alternative explanation of their results is that the checkpoint silencing activity is stronger on chromosomes. The APC/C pool bound to chromosomes is selectively activated first and preferably ubiquitinates chromosome-bound cyclin B1. This alternative model needs to be discussed, especially if the in vitro ubiquitination assay mentioned above cannot be done or produces negative results.

We thank the referee for this suggestion. We believe we have excluded this possibility by assaying Cyclin B1 degradation in the presence of Reversine to eliminate the MCC. We found that Cyclin B1 is readily degraded at NEBD, as expected, but CCNB1^{4E7E}-mEmerald^{+/+} degradation was delayed. We now include this result in figure S10D

(3) The authors show that RPE-1 CCNB14E7E-mEmerald^{+/+} cells became aneuploid. How many clones were examined? Can this phenotype be observed with multiple clones? Does the spindle checkpoint work properly in these cells?

Three independent clones were examined, and the SAC is fully functional. As outlined in response to referee 2, we think that the explanation is that the cells have too much Cyclin B1 in late metaphase/anaphase.

Dr. Jonathon Pines
The Institute of Cancer Research
237 Fulham Road
London
United Kingdom

11th Jul 2024

Re: EMBOJ-2024-116650R
Spatial control of the APC/C ensures the rapid degradation of Cyclin B1

Dear Jon,

Thank you again for submitting your revised manuscript to The EMBO Journal. It has been re-reviewed by all three original referees, whose comments are copied below. Following our pre-discussions of the remaining issues raised mainly by referee 2, I would invite you to prepare a formal point-by-point letter responding to/clarifying these concerns, and to incorporate additional explanations into the manuscript as needed. Where appropriate, you may also add some exemplary Expanded View movies (which should be uploaded combined with their legends in ZIP archives, and called Movie EV1/2/...) for further illustration. At this stage, please also take care of several remaining editorial issues that still need to be addressed:

- Naming conventions for main figures, Expanded View figures, and Appendix items: The 5 Expanded View Figures have to be renamed within the files, the legends, and throughout the text into "Figure EV1/2/3...". The Appendix figures have to be numbered starting again from 1, and have to be labelled "Appendix Figure S1/2/3..." both inside the Appendix and when referring them in the main text (calling out e.g. "Fig S1G" would not be specific enough). Tables in the Appendix also have to always be labelled as "Appendix Table S1/2/3" not just Table S1, also when calling them out in the text. Finally, please reference the article's title and authors on the first page of the Appendix PDF.
- On the abstract page of the manuscript, please include 4-5 general keyword terms to enhance searchability.
- Please rename the conflict of interest statement into "Disclosure and competing interests statement" as specified in our Guide to Authors (see <https://www.embopress.org/competing-interests> for details).
- As we are switching from a free-text author contribution statement towards a more formal statement based on Contributor Role Taxonomy (CRediT) terms, please remove the present Author Contribution section and instead specify each author's contribution(s) directly in the Author Information page of our submission system during upload of the final manuscript. See <https://casrai.org/credit/> for more information.
- In the Data Availability section, please include (spell out) hyperlinks to the specific databases (e.g., PDB, PXD) in which data have been deposited. Also, please remove the referee access tokens now, and ensure that the data will be promptly released latest at the time of online publication.
- Please remove the hyperlinks from the reference list (please refer to our Guide to Authors for additional information on EMBO J reference format).
- We had previously noted that error bars were not defined in the legends of figures 2b; 4c, i-j; 5a-b; 6c-d; 7b-d. Although you indicated that this is explained in the legend for figure 1, it needs to be explicitly stated in each individual figure legend.
- Finally, please provide suggestions for a short 'blurb' text prefacing and summing up the study in two sentences (max. 250 characters), followed by 3-5 one-sentence 'bullet points' with brief factual statements of key results of the paper; they will form the basis of an editor-written 'Synopsis' accompanying the online version of the article. Please also upload a synopsis image, which can be used as a "visual title" for the synopsis section of your paper. The image should be in PNG or JPG format with the modest dimensions of EXACTLY 550 pixels wide and 300-600 pixels high.

I am therefore returning the manuscript to you for a final round of minor revision, to allow you to make these adjustments and clarifications, and to upload all modified files. Once we will have received them, we should hopefully be able to swiftly proceed with formal acceptance and production of the manuscript.

Yours sincerely,

Hartmut

*** PLEASE NOTE: All revised manuscripts are subject to initial checks for completeness and adherence to our formatting guidelines. Revisions may be returned to the authors and delayed in their editorial re-evaluation if they fail to comply to the following requirements (see also our Guide to Authors for further information):

9) Digital image enhancement is acceptable practice, as long as it accurately represents the original data and conforms to community standards. If a figure has been subjected to significant electronic manipulation, this must be clearly noted in the figure legend and/or the 'Materials and Methods' section. The editors reserve the right to request original versions of figures and the original images that were used to assemble the figure. Finally, we generally encourage uploading of numerical as well as gel/blot image source data; for details see: embopress.org/page/journal/14602075/authorguide#sourcedata

At EMBO Press, we ask authors to provide source data for the main manuscript figures. Our source data coordinator will contact you to discuss which figure panels we would need source data for and will also provide you with helpful tips on how to upload and organize the files.

In the interest of ensuring the conceptual advance provided by the work, we recommend submitting a revision within 3 months (9th Oct 2024). Please discuss the revision progress ahead of this time with the editor if you require more time to complete the revisions. Use the link below to submit your revision:

Link Not Available

Referee #1:

The authors have largely addressed my concerns with either text or experimental changes.

However, in order to address the concern about the APC3 loop interacting with the nucleosome, the authors performed two EMSAs either with an APC3 Loop mutant or LANA peptide. In both experiments, the APC3 Loop-NPC complex band does not appear when the LANA peptide or variant loop is present, but the NCP147 bands seem to still disappear, indicating that some level of binding is intact (Figure 3D-E). Therefore, perhaps these data can be moved to the supplement or omitted as the new cryo-EM structure (Figure 3F-G) strongly supports the APC3 loop interaction with the acidic patch.

Referee #2:

The revised manuscript by Cirillo et al. endeavors to elucidate the molecular mechanisms that govern the spatial regulation of Cyclin B1 degradation via its interaction with nucleosomes during mitosis. The addition of new data enhances our understanding of the role of Cyclin B1's N-terminal acidic patch in nucleosome binding and its impact on degradation efficiency. The novel observation of Cyclin B1 directly interacting with nucleosomes is intriguing and marks a significant advance in our understanding of mitotic regulation.

However, the manuscript does not adequately address several critical areas. Despite improvements, evidence supporting the interaction between APC/C subunits and nucleosomes remains predominantly *in vitro*. Additionally, the functional relevance of these interactions for Cyclin B1 degradation is not sufficiently substantiated, and the physiological implications of these interactions for genomic stability are not convincingly demonstrated. The organization of the manuscript also requires improvement to better convey the core findings.

In summary, while the manuscript provides valuable insights into the role of the N-terminal acidic patch and nucleosome binding of mammalian Cyclin B for controlled degradation during mitotic exit, it falls short in fully substantiating its central claims, particularly regarding the functional importance of APC/C interactions with nucleosomes.

Specific comments:

1. The manuscript claims that direct interactions between APC/C and nucleosomes are crucial for Cyclin B1 degradation. This assertion is innovative and significant, suggesting a novel layer of regulation in mitotic control. However, the evidence provided—primarily derived from *in vitro* experiments using purified components, short fragments, and fluorescence cross-correlation spectroscopy (FCCS) focusing only on the APC8 subunit—does not convincingly support these interactions *in vivo* or within the context of the entire APC/C complex. The study lacks functional analysis of nucleosome-binding deficient APC3 mutants in cellular settings, which would be crucial to validate the proposed mechanisms.
2. The documentation of Apc4 immuno-depletion in new Figure S1E suggests some incorporation of tagged Apc8 into APC/C, but it does not confirm complete integration for all the tagged APC8, as approximately 50% of Apc4 remains post-depletion. Nearly complete depletion is generally expected in such assays. Analogous analysis with other tagged APC/C subunits, which the authors have generated, would help clarify this aspect further.
3. Size exclusion chromatography data presented in Figure 6A indicate that Cyclin B acidic patch mutants do not markedly affect the APC/CCDC20-Cyclin B interaction. Nonetheless, this does not assess the first nine amino acids' impact on ubiquitination efficiency. Given that APC/C/CCDC20-binding and ubiquitination are linked but separable processes, the experiment using the LANA motif (Figure 6C) suggests that nucleosome interaction promotes substrate degradation. However, a comparison between LANA-fused wild type Cyclin B and LANA-Cyclin B1 Δ 9 is necessary to confirm if they show comparable degradation kinetics.
4. Although posited as crucial for genomic stability, the interactions between Cyclin B1 and nucleosomes leading to timely Cyclin B degradation need further validation. The long-term effects of Cyclin B1 mutants on cell ploidy are predictable given the importance of cyclin degradation in cell cycle control. A more nuanced approach, such as detailed analyses of mitotic progression in Cyclin B1 mutants disrupting nucleosome binding, would provide deeper insights into the mechanisms causing aneuploidy.
5. The inaccessibility of source data files has limited the ability to verify the statistical analyses and normalization procedures of quantification data. Ensuring easy access to these files and providing detailed methodological explanations would greatly aid in the reproducibility and validation of the results.
6. Optimizing the organization of the manuscript, particularly the arrangement of data between main figures and supplementary figures, could significantly enhance narrative clarity. Inclusion of dynamic representations such as movies could further substantiate the dynamic processes discussed.
7. Additional references and corrections are needed to enhance accuracy and clarity, including citing Fujimitsu et al., 2016 in Line 451, correcting the label in Figure 3E to "APC3 Loop3R3E," removing duplicated texts in Figure S4C, and ensuring consistency in Cyclin B1 nomenclature across figures.

Referee #3:

The authors have addressed two of my concerns. They also made a good effort in attempting to address my first concern. Although they were not successful in demonstrating the stimulatory effect of nucleosomes in vitro, they provided reasonable explanations for the failure. The discussed mechanisms are plausible. Overall, I can now support the publication of this revised manuscript.

Responses to Referees:

Referee #1

The authors have largely addressed my concerns with either text or experimental changes.

However, in order to address the concern about the APC3 loop interacting with the nucleosome, the authors performed two EMSAs either with an APC3 Loop mutant or LANA peptide. In both experiments, the APC3 Loop-NPC complex band does not appear when the LANA peptide or variant loop is present, but the NCP147 bands seem to still disappear, indicating that some level of binding is intact (Figure 3D-E). Therefore, perhaps these data can be moved to the supplement or omitted as the new cryo-EM structure (Figure 3F-G) strongly supports the APC3 loop interaction with the acidic patch.

We agree with this referee that both the wt and mutant APC3 loop can shift the nucleosome. This is probably due to non-specific binding of the APC3 loop with the nucleosome; however, only the wild type and not the mutant loop can generate a well-defined specific shifted band, which is the sign of complex formation. We will add the sentence above to the manuscript.

Referee #2 (Report for Author)

The revised manuscript by Cirillo et al. endeavors to elucidate the molecular mechanisms that govern the spatial regulation of Cyclin B1 degradation via its interaction with nucleosomes during mitosis. The addition of new data enhances our understanding of the role of Cyclin B1's N-terminal acidic patch in nucleosome binding and its impact on degradation efficiency. The novel observation of Cyclin B1 directly interacting with nucleosomes is intriguing and marks a significant advance in our understanding of mitotic regulation.

However, the manuscript does not adequately address several critical areas. Despite improvements, evidence supporting the interaction between APC/C subunits and nucleosomes remains predominantly *in vitro*. Additionally, the functional relevance of these interactions for Cyclin B1 degradation is not sufficiently substantiated, and the physiological implications of these interactions for genomic stability are not convincingly demonstrated. The organization of the manuscript also requires improvement to better convey the core findings.

In summary, while the manuscript provides valuable insights into the role of the N-terminal acidic patch and nucleosome binding of mammalian Cyclin B for controlled degradation during mitotic exit, it falls short in fully substantiating its central claims, particularly regarding the functional importance of APC/C interactions with nucleosomes.

Specific comments:

1. The manuscript claims that direct interactions between APC/C and nucleosomes are crucial for Cyclin B1 degradation. This assertion is innovative and significant, suggesting a novel layer of regulation in mitotic control. However, the evidence provided—primarily derived from *in vitro* experiments using purified components, short fragments, and fluorescence cross-correlation spectroscopy (FCCS) focusing only on the APC8 subunit—does not convincingly support these interactions *in vivo* or within the context of the entire APC/C complex. The study lacks functional analysis of nucleosome-binding deficient APC3 mutants in cellular settings, which would be crucial to validate the proposed mechanisms.

This criticism is incorrect. We conclude that Cyclin B1 interaction with nucleosomes is crucial for its degradation, but we do not claim that the APC/C interaction is crucial: in our Discussion we state that our data indicate that binding is important. We intend to generate an APC3 arginine anchor mutant but since this could also alter its nuclear import, these studies will need to be carefully executed and interpreted and thus belong in a follow up study.

With respect to the evidence that the APC/C bind nucleosomes, there are supporting data from Skranja et al (doi.org/10.1093/nar/gkaa544) who showed that the APC/C binds to nucleosomes in an acidic patch-dependent manner. We have improved on these data by elucidating the mechanism of binding (through the APC3 loop binding directly to the acidic patch).

2. The documentation of Apc4 immuno-depletion in new Figure S1E suggests some incorporation of tagged Apc8 into APC/C, but it does not confirm complete integration for all the tagged APC8, as approximately 50% of Apc4 remains post-depletion. Nearly complete depletion is generally expected in such assays. Analogous analysis with other tagged APC/C subunits, which the authors have generated, would help clarify this aspect further.

We take issue with this interpretation of our data. Our experiment shows that immunoprecipitating the APC/C with an anti-APC4 antibody brings down the same proportion of APC8 regardless of whether this is from untagged or our tagged cell line. Thus, tagging APC8 does not alter its behaviour. Furthermore, we back up these data by FCS, which shows APC8-mScarlet diffuses as a single, high molecular weight species in living cells (Figure S1G). As we mentioned in our previous response, APC1, APC3 and APC4 tagged cell lines all show the same behaviour as tagged-APC8. Lastly, mitosis and Cyclin B1 degradation are indistinguishable in the tagged and untagged cells.

3. Size exclusion chromatography data presented in Figure 6A indicate that Cyclin B acidic patch mutants do not markedly affect the APC/CCDC20-Cyclin B interaction. Nonetheless, this does not assess the first nine amino acids' impact on ubiquitination efficiency. Given that APC/C/CDC20-binding and ubiquitination are linked but separable processes, the experiment using the LANA motif (Figure 6C) suggests that nucleosome interaction promotes substrate degradation. However, a comparison between LANA-fused wild type Cyclin B and LANA-Cyclin B1 Δ 9 is necessary to confirm if they show comparable degradation kinetics.

We don't see the necessity for this experiment. As the referee says, the LANA peptide shows that nucleosome interaction promotes degradation, which was the point of our experiment.

4. Although posited as crucial for genomic stability, the interactions between Cyclin B1 and nucleosomes leading to timely Cyclin B degradation need further validation. The long-term effects of Cyclin B1 mutants on cell ploidy are predictable given the importance of cyclin degradation in cell cycle control. A more nuanced approach, such as detailed analyses of mitotic progression in Cyclin B1 mutants disrupting nucleosome binding, would provide deeper insights into the mechanisms causing aneuploidy.

We don't think this is a reasonable criticism. In their original critique the referee asked us to perform an induction study, which we have done. This revealed that failure of cytokinesis is a major contributor to aneuploidy, which we discuss. Moreover, the referee asserts that the long-term effects of Cyclin B1 mutants on cell ploidy are predictable without citing any evidence to back up this claim. Given the multiple roles of Cyclin B1-CDK1 in the cell cycle, we would not have the confidence to predict the effect of delayed degradation ourselves. Nevertheless, it is implicit in the referee's remarks that they agree that timely degradation of Cyclin B1 degradation is crucial for genomic stability.

5. The inaccessibility of source data files has limited the ability to verify the statistical analyses and normalization procedures of quantification data. Ensuring easy access to these files and providing detailed methodological explanations would greatly aid in the reproducibility and validation of the results.

We apologise for our oversight and provide the correct Source data file as requested.

6. Optimizing the organization of the manuscript, particularly the arrangement of data between main figures and supplementary figures, could significantly enhance narrative clarity. Inclusion of dynamic representations such as movies could further substantiate the dynamic processes discussed.

We have discussed with the editor the possibility of including movies and include two representative movies of Cyclin B1 degradation.

7. Additional references and corrections are needed to enhance accuracy and clarity, including citing Fujimitsu et al., 2016 in Line 451, correcting the label in Figure 3E to "APC3 Loop3R3E," removing duplicated texts in Figure S4C, and ensuring consistency in Cyclin B1 nomenclature across figures.

We have rectified these points.

Referee #3 (Report for Author)

The authors have addressed two of my concerns. They also made a good effort in attempting to address my first concern. Although they were not successful in demonstrating the stimulatory effect of nucleosomes in vitro, they provided reasonable explanations for the failure. The discussed mechanisms are plausible. Overall, I can now support the publication of this revised manuscript.

We thank the referee for their positive evaluation. We are currently working on in vitro ubiquitylation assays and we hope to present these in future work.

Dr. Jonathon Pines
The Institute of Cancer Research
237 Fulham Road
London
United Kingdom

26th Jul 2024

Re: EMBOJ-2024-116650R1
Spatial control of the APC/C ensures the rapid degradation of Cyclin B1

Dear Jon,

Thank you for submitting your final revised manuscript for our consideration. I am pleased to inform you that we have now accepted it for publication in The EMBO Journal.

With kind regards,

Hartmut
